# TranSUN: A Preemptive Paradigm to Eradicate Retransformation Bias Intrinsically from Regression Models in Recommender Systems

**Jiahao Yu**$^{*\dagger}$**, Haozhuang Liu**$^{*}$**, Yeqiu Yang**$^{*}$**, Lu Chen, Jian Wu, Yuning Jiang, Bo Zheng**

Taobao & Tmall Group of Alibaba, Beijing, China

{yujiahao.yjh,liuhaozhuang.lhz,yangyeqiu.yyq,chenyou.cl,
joshuawu.wujian,mengzhu.jyn,bozheng}@alibaba-inc.com

## Abstract

Regression models are crucial in recommender systems. However, retransformation bias problem has been conspicuously neglected within the community. While many works in other fields have devised effective bias correction methods, all of them are *post-hoc* cures *externally* to the model, facing practical challenges when applied to real-world recommender systems. Hence, we propose a *preemptive* paradigm to eradicate the bias *intrinsically* from the models via minor model refinement. Specifically, a novel **TranSUN** method is proposed with a joint bias learning manner to offer theoretically guaranteed unbiasedness under empirical superior convergence. It is further generalized into a novel generic regression model family, termed **G**eneralized **T**ranSUN (**GTS**), which not only offers more theoretical insights but also serves as a generic framework for flexibly developing various bias-free models. Comprehensive experimental results demonstrate the superiority of our methods across data from various domains, which have been successfully deployed in two real-world industrial recommendation scenarios, *i.e.* product and short video recommendation scenarios in *Guess What You Like* business domain in the homepage of Taobao App (a leading e-commerce platform with DAU > 300M), to serve the major online traffic.

## 1 Introduction

Regression models play a pivotal role in recommender systems, which have been widely applied to prediction tasks for targets including user engagement duration [38, 9, 23, 22], transaction amount [15, 42, 44], lifetime value [26, 41, 40, 21, 24], *etc*. Mean squared error (MSE) loss is a predominant choice for loss functions of regression models, which adopts for target $y$ and features $x$ an apriori assumption of $y|x \sim \mathcal{N}(f(x;\theta), \sigma^2)$, where $\mathcal{N}(f(x;\theta), \sigma^2)$ is a Gaussian distribution with $f(x;\theta)$ as mean and $\sigma^2$ as variance and $f(x;\theta)$ is a $\theta$-parameterized predictor aiming to learn $\mathbb{E}_{y\sim\mathcal{P}(Y|x)}[y]$. However, the real target distribution frequently violates the Gaussian assumption in recommendation scenarios due to the high-skewed characteristics of the target data [7, 41, 40], which critically impedes the model convergence. This problem motivates prevalent adoption of various transformations T for target $y$ in the MSE-based regression models, *abbr.* transformed MSE, which facilitates the conformity between the transformed distribution $\mathcal{P}(\mathrm{T}(Y)|x)$ and the hypothetical distribution $\mathcal{N}(f(x;\theta), \sigma^2)$, thereby improving the performance of model convergence [32, 31].

Though effective, it leads to the *retransformation bias* problem [2, 31, 37], *i.e.* the estimate for $\mathbb{E}_{y\sim\mathcal{P}(Y|x)}[y]$ is biased when the prediction $f(x;\theta)$ of the transformed MSE model is back-

---

$^{*}$These authors contributed equally to this research.

$^{\dagger}$Corresponding author.

39th Conference on Neural Information Processing Systems (NeurIPS 2025).

transformed to the original space, as

$$\mathrm{T}^{-1}\big(f(x;\theta)\big) \neq \mathbb{E}_{y\sim\mathcal{P}(\mathrm{Y}|x)}[y], \tag{1}$$

where $\mathrm{T}^{-1}$ is the inverse function of $\mathrm{T}$. This critical issue has been conspicuously neglected within the community of recommender systems. Fortunately, there are still many effective works in other fields specifically studying the bias problem, all of which resort to the statistics of the model residuals to design post-processing bias correction mechanisms exactly after the model has been trained [12, 13, 33, 31, 14]. Despite the progress, all the prior works are *post-hoc* approaches designed and applied *externally* to the model, which face practical challenges when applied to the real-world recommender systems (detailed in Appendix C). Therefore, we challenge the prior paradigm by asking: Can we eradicate the retransformation bias *intrinsically* from the model in a *preemptive* manner?

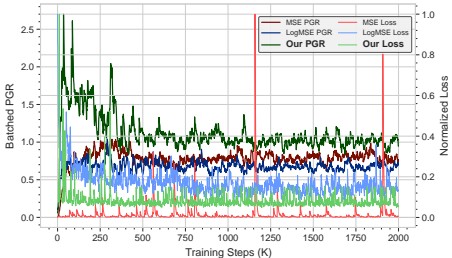

Figure 1: Illustration of biasedness and convergence of MSE, LogMSE, and our TranSUN ($\mathrm{T}$ is logarithmic) during the training on the Indus dataset, where PGR denotes the ratio of the prediction mean to the ground truth mean in a batch and the loss value is normalized by the maximum value. As illustrated, MSE model has difficulty converging, while LogMSE model is significantly biased (*i.e.* PGR < 1.0). Differently, our method simultaneously maintains prediction unbiasedness (*i.e.* PGR is near 1.0) and superior convergence.

To answer this, this paper proposes a novel preemptive paradigm to eradicate bias from the model exactly during the training phase via minor model refinement. Specifically, we propose the novel **TranSUN** method to empower the **trans**formed MSE model with intrinsic **un**biasedness via introducing a jointly-learned auxiliary model branch for explicit bias modeling. Our method not only achieves theoretically guaranteed unbiasedness but also exhibits superior empirical convergence, as shown in Fig. 1. Interestingly, we point out that not all explicit bias modeling schemes bring theoretically guaranteed unbiasedness, and we further generalize TranSUN into a novel generic regression model family, termed **G**eneralized **T**ran**SUN** (**GTS**), to shed light on the underlying mechanism to intrinsically maintain model's unbiasedness. Essentially, GTS establishes the intrinsic unbiasedness through exploiting target transformation with *conditional linearity* based on particular point estimates [20] of $\mathrm{T}(y)|x$, exhibiting a more general model assumption compared to the vanilla MSE model. With this generic framework, not only various bias-free regression models can be directly devised to cope with varying data scenarios, but also any given regression model can be flexibly debiased in a plug-and-play manner even if the given model does not utilize any target transformation.

The major contributions can be summarized as follows.

- We propose a novel *preemptive* paradigm to eradicate retransformation bias *intrinsically* from regression models to challenge the conventional ones, which is also the first work in the recommender system community to address the long-neglected bias problem.

- We propose the novel **TranSUN**, a theoretically grounded, empirical convergence-friendly, and transformation-agnostic methodology, to endow the **trans**formed MSE model with intrinsic **un**biasedness exactly during the training phase.

- We further propose a novel regression model family, termed **G**eneralized **T**ran**SUN** (**GTS**), to offer more theoretical insights, which also provides a generic framework for flexibly developing various bias-free regression models. Notably, our TranSUN and GTS have been validated to be directly applicable to other fields beyond the recommender system domain.

- Extensive experiments on synthetic and real-world data validate the superiority and broad applicability of our methods. Now, TranSUN has been deployed in two core recommendation scenarios, *i.e.* product and short video recommendation, in *Guess What You Like* domain in the homepage of Taobao App (a leading e-commerce platform with DAU > 300M) to serve the major online traffic.

## 2 Related works

**Regression models in recommender systems** mainly fall into two categories, *i.e.* value regression and classification-based regression, and our method belongs to the former. Value regression defines target as a continuous random variable and directly predicts target value using the expectation of the hypothetical distribution [22, 7, 48, 41, 42, 44], *e.g.* Gaussian distribution adopted by the MSE-based model family [42, 44, 43], ZILN distribution [40, 41]. By contrast, classification-based regression defines target as a discrete random variable via dividing it into multiple bins [45, 46, 9, 47], which mainly introduces either hierarchical routing [23, 21, 24, 39] or ordinal dependencies [23, 38] to boost models. Despite the progress, all the prior works neglect the study of model biasedness problems (*e.g.* retransformation bias problem), which probably lead to the systematic errors in the model predictions (as shown in Eq. (3)), *e.g.* systematic underestimation of duration models in video recommender systems. Differently, our paper specifically addresses this critical biasedness problem.

**Retransformation bias** was initially identified in [2], with several works deriving theoretical correction terms by assuming the transformed distribution [13, 3, 8, 31]. Subsequent studies advanced correction frameworks free from distributional assumptions [12, 35, 5, 37, 27], achieving notable success across various fields [11, 19, 18, 30, 25, 1, 14, 34, 36, 33, 37]. Nonetheless, these approaches are still post-hoc remedies out of the model, bringing practical challenges when applied to industrial recommender systems. Differently, we preemptively address the bias via intrinsic model refinement.

**Conditional transformation models (CTM)** learns a parametric transformation introducing interaction between target $y$ and input $x$ to transform them into variables of assumed distributions [16, 28, 4]. CLTM is further restricted to using affine transformation for enhanced interpretability [29], whose model assumption is $\alpha(x) + h_0(Y_x) \cdot \beta(x) \sim \mathcal{N}(0,1)$ with $Y_x$ denoting $y|x$ and $h_0(\cdot)$ being parameter-free. Though CLTM and our GTS are superficially similar, they have one key differences, *i.e. different dependencies* in model assumptions, where the slope $\beta(x)$ in CLTM's is only related to $x$, while ours (Eq. (11)) is related to both $x$ and $y$. In addition, regarding model classes and motivations, CLTM belongs to transformation models [17] for interval estimation, while GTS is a value regression model for unbiased point estimation.

## 3 Method

### 3.1 Preliminary

Given a training set $\mathcal{D} = \{(x,y)|(x,y) \sim \mathcal{P}(X,Y)\}$, where $x$ denotes the input features and $y$ denotes the continuous target label, the goal of the regression task is to make the model prediction $f(x;\theta)$ and the ground truth $y$ as close as possible, where $\theta$ is the parameter of the model $f$. In the recommender system community, a transformed MSE model is commonly adopted to boost the model convergence performance on high-skewed data, which is formulated as

$$\mathcal{L}_{\text{MSE}}^{\text{T}} = \mathbb{E}_{(x,y)\sim\mathcal{D}}\Big[\big(f(x;\theta) - \text{T}(y)\big)^2\Big], \tag{2}$$

where T is a customized bijective mapping. However, since minimizing Eq. (2) makes $f(x;\theta)$ be an estimate of $\mathbb{E}_{y\sim\mathcal{P}(Y|x)}\big[\text{T}(y)\big]$ (derived in Appendix B.1), directly back-transforming $f(x;\theta)$ leads to the retransformation bias problem, as shown in Eq. (1). Taking a common example in the industrial community, assuming the inverse transformation $\text{T}^{-1}$ is convex, *e.g.* $\text{T}(y)$ is $\log(y+1)$ or $\sqrt{y}$, then the retransformation bias will result in the systematic underestimation of model prediction $\hat{y}|x$, as

$$\hat{y}|x = \text{T}^{-1}(f(x;\theta)) = \text{T}^{-1}\Big(\mathbb{E}_{y\sim\mathcal{P}(Y|x)}\big[\text{T}(y)\big]\Big) \leq \mathbb{E}_{y\sim\mathcal{P}(Y|x)}\Big[\text{T}^{-1}\big(\text{T}(y)\big)\Big] = \mathbb{E}_{y\sim\mathcal{P}(Y|x)}[y], \tag{3}$$

where $\leq$ is obtained by Jensen's inequality and it is reversed when $\text{T}^{-1}$ is concave.

### 3.2 TranSUN: joint bias learning

To intrinsically eradicate the retransformation bias, we propose TranSUN, which introduces an auxiliary branch $z(x;\theta_z)$ to the transformed MSE model to explicitly learn the bias. Specifically,

we employ a multiplicative bias modeling scheme, *i.e.* supervising $z(x; \theta_z)$ to learn the ratio of the ground truth $y$ to the biased prediction $\mathrm{T}^{-1}(f(x; \theta))$, hence the bias learning loss is formulated as

$$\mathcal{L}_{\text{sun}}^{\mathrm{T}} = \mathbb{E}_{(x,y)\sim\mathcal{D}}\Big[\Big(z(x; \theta_z) - \text{stop\_grad}\Big[\frac{y}{|\mathrm{T}^{-1}(f(x; \theta))| + \epsilon}\Big]\Big)^2\Big], \tag{4}$$

where $\text{stop\_grad}$ is applied to prevent the gradients of $\mathcal{L}_{\text{sun}}^{\mathrm{T}}$ from passing to $\theta$, $|x|$ denotes the absolute value of $x$, and $\epsilon$ is a positive hyperparameter for avoiding division by zero. The reason for adopting this scheme is that the target ratio has intuitively small variance, empirically making the loss much smoother (as shown in Fig. 3a and discussed in Appendix A.2). During training, our model jointly learns the bias modeling task and the main regression task to achieve preemptive debiasing, thus the total loss is formulated as

$$\mathcal{L}_{\text{TranSUN}}^{\mathrm{T}} = \mathcal{L}_{\text{MSE}}^{\mathrm{T}} + \mathcal{L}_{\text{sun}}^{\mathrm{T}}. \tag{5}$$

In the testing phase, the model prediction $\hat{y}|x$ is formulated as

$$\hat{y}|x = z(x; \theta_z) \cdot \big(|\mathrm{T}^{-1}(f(x; \theta))| + \epsilon\big). \tag{6}$$

**Theoretical analysis of unbiasedness.** To illustrate the inherent unbiasedness of our method, we prove the prediction $\hat{y}|x$ to be an estimate of $\mathbb{E}_{y\sim\mathcal{P}(\mathrm{Y}|x)}[y]$ as follows (see Appendix B.2 for details). Owing to the $\text{stop\_grad}$ operation, the optimization of the total loss $\mathcal{L}_{\text{TranSUN}}^{\mathrm{T}}$ can be decomposed into two independent parts, as

$$\min_{\theta, \theta_z} \mathcal{L}_{\text{TranSUN}}^{\mathrm{T}} \Leftrightarrow \min_{\theta} \mathcal{L}_{\text{MSE}}^{\mathrm{T}} + \min_{\theta_z} \mathcal{L}_{\text{sun}}^{\mathrm{T}}. \tag{7}$$

In the second part, the prediction $\hat{y}|x$ is derived as the estimate of $\mathbb{E}_{y\sim\mathcal{P}(\mathrm{Y}|x)}[y]$ via

$$\min_{\theta_z} \mathcal{L}_{\text{sun}}^{\mathrm{T}} \Leftrightarrow \forall x \sim \mathcal{P}(\mathrm{X}), \ \mathbb{E}_{y\sim\mathcal{P}(\mathrm{Y}|x)}\Big[z(x; \theta_z) - \frac{y}{|\mathrm{T}^{-1}(f(x; \theta))| + \epsilon}\Big] = 0$$
$$\Leftrightarrow z(x; \theta_z) \cdot \big(|\mathrm{T}^{-1}(f(x; \theta))| + \epsilon\big) = \mathbb{E}_{y\sim\mathcal{P}(\mathrm{Y}|x)}[y] \ \Leftrightarrow \ \hat{y}|x = \mathbb{E}_{y\sim\mathcal{P}(\mathrm{Y}|x)}[y]. \tag{8}$$

**Key discussion.** According to Eq. (8), we propose an interesting argument: *not all bias learning schemes provide theoretically guaranteed unbiasedness*. As an example, supposing we turn the learning target of $z(x; \theta_z)$ in Eq. (4) to the ratio of biased prediction to ground truth, *i.e.* $\mathrm{T}^{-1}(f(x; \theta))/y$, then the model prediction is derived as an estimate of $1/\mathbb{E}_{y\sim\mathcal{P}(\mathrm{Y}|x)}[\frac{1}{y}]$ (derived in Appendix B.3), which is a biased estimate bringing model underestimation according to Eq. (3). More empirical results are shown in Table 5. Therefore, in the next section, we point out that it is *conditional linearity* as the essential mechanism for the unbiasedness of TranSUN, rather than the explicit bias learning.

### 3.3 Generalized TranSUN (GTS)

In this section, we generalize TranSUN to a novel generic regression model family, termed Generalized TranSUN (GTS), to provide more insights into the intrinsic unbiasedness mechanism of our method. The total loss function of GTS is formulated as

$$\mathcal{L}_{\text{GTS}} = \mathcal{L}_{\mathcal{H}_q}\Big(x, \mathrm{T}(y); \theta_q\Big) + \mathbb{E}_{(x,y)\sim\mathcal{D}}\Big[\Big(z(x; \theta_z) - y \cdot \text{stop\_grad}\big[\kappa\big(f(x; \theta_q)\big)\big]\Big)^2\Big], \tag{9}$$

where $\mathcal{H}_q$ is a known apriori distribution parameterized by $x$ and $\theta_q$, $f(x; \theta_q)$ is the expectation of the hypothesized distribution $\mathcal{H}_q(x, \theta_q)$, and $\kappa$ is a parameter-free function with $\kappa(\cdot) > 0$.

Specifically, the first loss term $\mathcal{L}_{\mathcal{H}_q}$ is termed *conditional point loss*, which supervises $f(x; \theta_q)$ to specifically learn a conditional point estimate $\mathbb{Q}_{y\sim\mathcal{P}(\mathrm{Y}|x)}\big[\mathrm{T}(y)\big]$ (*e.g.* arithmetic mean, median, mode) via customizing the apriori conditional probability assumption of $\mathrm{T}(y)|x \sim \mathcal{H}_q(x, \theta_q)$, with

$$f(x; \theta_q) = \mathbb{E}_{\mathrm{T}(y)\sim\mathcal{H}_q(x,\theta_q)}[\mathrm{T}(y)] = \mathbb{Q}_{\mathrm{T}(y)\sim\mathcal{P}(\mathrm{T}(\mathrm{Y})|x)}\big[\mathrm{T}(y)\big] = \mathbb{Q}_{y\sim\mathcal{P}(\mathrm{Y}|x)}\big[\mathrm{T}(y)\big]. \tag{10}$$

The second loss term in Eq. (9) is termed *linear transformation loss*, which much resembles linear-transformed MSE, whereas the key difference is that our slope $\kappa\big(f(x; \theta_q)\big)$ is dynamically generated by the learned conditional point $f(x; \theta_q)$. Additionally, due to the $\text{stop\_grad}$ operation, the slope $\kappa\big(f(x; \theta_q)\big)$ is actually completely determined by $\mathcal{L}_{\mathcal{H}_q}$. In the testing phase, the prediction of GTS is $\hat{y}|x = z(x; \theta_z)/\kappa\big(f(x; \theta_q)\big)$, which can also be proved unbiased by following the derivation process in Eq. (7, 8) (also derived in Appendix B.4).

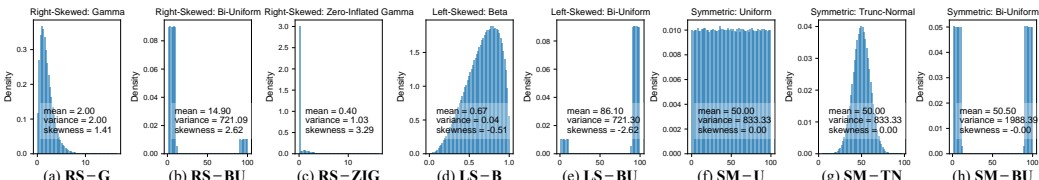

Figure 2: Density and statistics of the sampled $\mathcal{P}_s(Y|x)$ from the eight pre-defined $\mathcal{H}(Y)$.

**Model assumption.** Given any sample $x$, GTS essentially exploits a customized $x$-conditioned point (*abbr.* $\mathbb{Q}$) to generate a $\mathbb{Q}$-conditioned slope $\kappa$ for linear transformation of target $y$. Hence, the model assumption of GTS is formulated as

$$-z(x;\theta_z) + Y_x \cdot \kappa\big(\mathbb{Q}_{y\sim\mathcal{P}(Y|x)}\big[\mathrm{T}(y)\big]\big) \sim \mathcal{N}(0,\sigma^2), \tag{11}$$

where $Y_x = (y|x)$ and $\mathcal{N}$ denotes Gaussian distribution. As shown in Eq. (11), the slope $\kappa$ is conditioned on $\mathcal{P}(Y|x)$, making GTS akin to a piecewise-linear-transformed MSE where the intervals are determined by the condition $x$. Accordingly, GTS can be viewed as a *conditional* linear-transformed MSE, the unbiasedness of which is naturally maintained since linear-transformed MSE is unbiased itself. Notably the violation of model assumptions will not invalidate the model's theoretical unbiasedness, which only affects the model performance (*e.g.* convergence). Furthermore, according to Eq. (11), the variance of the hypothetical distribution of $y|x$ is modeled as $\sigma^2/\kappa(\mathbb{Q})$, where one can explicitly model the heteroscedasticity by specifying $\mathbb{Q}$ and $\kappa$ with the domain knowledge, exhibiting a more general model assumption than the traditional MSE model.

**Application.** There are two main application scenarios for the proposed GTS. The first is to directly devise various *bias-free* regression models by customizing the assumption $\mathrm{T}(y)|x \sim \mathcal{H}_q(x,\theta_q)$ and $\kappa$'s function type to cope with different data scenarios. Some design guidelines are shown in Appendix A.6, with reference results provided in Table 15, 10. The second is to serve as a plug-and-play *debiasing* module to directly eradicate bias from any given regression model, even if the given model does not use any transformation $\mathrm{T}$. Specifically, for any given regression model $\mathcal{M}$, one can easily establish the debiased model $\mathcal{M}'$ by setting $\mathrm{T}$ to identity transformation, $\mathcal{L}_{\mathcal{H}_q}$ to the loss function of $\mathcal{M}$, and $f(x;\theta_q)$ to the prediction of $\mathcal{M}$. More empirical results are shown in Table 6.

## 4 Experiments on synthetic data

In this section, we conduct experiments on synthetic data to evaluate model performance. Specifically, we sample abundant instances $\{y_i\}_{i=1}^N$ from the pre-defined distribution $\mathcal{H}(Y)$ to form the conditional distribution $\mathcal{P}_s(Y|x)$, and then models are trained on $\{y_i\}_{i=1}^N$ for learning regression task, after which the error between model prediction and the theoretical-calculated expectation $\mathbb{E}_{y\sim\mathcal{H}(Y)}[y]$ are computed for evaluation.

### 4.1 Experiment settings

**Data and metric.** We adopt 8 distributions $\mathcal{H}(Y)$ from three categories: right-skewed, left-skewed, and symmetric. Details are shown in Appendix D.1.1. Fig. 2 illustrates the density and some statistics of the sampled $\mathcal{P}_s(Y|x)$. We adopt SRE (Signed Relative Error) as the metric, *i.e.* $(\hat{y} - y)/y$ where $\hat{y}$ and $y$ are prediction and ground truth, respectively.

**Implementation details.** See Appendix D.1.2 for the detailed introduction.

### 4.2 Results on synthetic data

**Effectiveness on eradicating retransformation bias.** Three transformations, *i.e.* linear, square, and logarithmic transformations, are employed to compare the performance between transformed MSE and our TranSUN. As demonstrated in Table 1, the retransformation bias problem arises in transformed MSE when the transformation is not linear, *i.e.* model becoming underestimated with $\mathrm{T}(y) = \ln(y + 1)$ and overestimated with $\mathrm{T}(y) = y^2$, which is consistent with Eq. (3). In contrast, TranSUN maintains unbiasedness ($|\mathrm{SRE}| < 0.7\%$) regardless of the transformation applied. Notably

Table 1: Comparison results of transformed MSE (T-MSE) and TranSUN under different transformations T on synthetic data. Red boldface indicates that **|SRE| >1%**. The difference to T-MSE is statistically significant at 0.02 level.

| Methods | $T(\cdot)$ | RS-G | RS-BU | RS-ZIG | LS-B | LS-BU | SM-U | SM-TN | SM-BU |
|---|---|---|---|---|---|---|---|---|---|
| MSE | - | 0.0010 | 0.0023 | 0.0015 | -0.0013 | -0.0022 | -0.0065 | 0.0005 | -0.0043 |
| T-MSE | $T(y) = 0.5y$ | -0.0012 | 0.0023 | -0.0013 | -0.0017 | -0.0024 | -0.0065 | -0.0003 | -0.0042 |
| **TranSUN** | | -0.0011 | 0.0022 | -0.0014 | -0.0015 | -0.0020 | -0.0066 | -0.0006 | -0.0045 |
| T-MSE | $T(y) = \ln(y+1)$ | **-0.1401** | **-0.5108** | **-0.4481** | **-0.0179** | **-0.1587** | **-0.2447** | **-0.0217** | **-0.5341** |
| **TranSUN** | | 0.0014 | 0.0013 | -0.0010 | 0.0014 | 0.0033 | -0.0038 | -0.0017 | -0.0058 |
| T-MSE | $T(y) = y^2$ | **0.2122** | **1.1015** | **1.7534** | **0.0479** | **0.0506** | **0.1442** | **0.0193** | **0.3264** |
| **TranSUN** | | -0.0045 | 0.0038 | 0.0019 | 0.0016 | 0.0063 | -0.0069 | -0.0035 | 0.0025 |

Table 2: Comparison results of advanced regression models on synthetic data. The **best** results are in boldface and the second best are underlined. Gray background indicates that |SRE| <1% .

| Methods | RS-G | RS-BU | RS-ZIG | LS-B | LS-BU | SM-U | SM-TN | SM-BU |
|---|---|---|---|---|---|---|---|---|
| MAE | -0.1508 | -0.5624 | -98.7456 | 0.0351 | 0.0967 | 0.0050 | **0.0006** | **0.0040** |
| WLR [9] | - | - | 0.2483 | - | - | - | - | - |
| ZILN [40] | -0.0032 | -0.2077 | -0.0022 | 0.0030 | 0.1866 | 0.1326 | 0.0024 | 0.2829 |
| MDME [21] | -0.1132 | -0.5876 | -0.1758 | 0.2730 | 0.1004 | 0.5500 | -0.1146 | -0.7511 |
| TPM [23] | 0.8303 | 0.1670 | 3.4244 | -0.0411 | -0.0303 | 0.0055 | 0.0080 | 0.0060 |
| CREAD [38] | 0.6511 | 0.1127 | 2.3508 | -0.0361 | -0.0291 | 0.0054 | 0.0060 | 0.0053 |
| OptDist [41] | -0.0024 | -0.2068 | -0.0021 | 0.0019 | 0.1869 | 0.1328 | 0.0022 | 0.2826 |
| **LogSUN** | **0.0014** | **0.0013** | **-0.0010** | **0.0014** | **0.0033** | **-0.0038** | -0.0017 | -0.0058 |

MSE and linear-transformed MSE both exhibit unbiasedness in Table 1, that is because they converge well due to the extreme simplicity of input features and model architectures. Differently, MSE exhibits significantly poor convergence on real-world data, as detailed in the next section (Table 7, 8).

**Comparison with state-of-the-arts.** Our TranSUN is further compared with state-of-the-art regression models in the recommender system field. As shown in Table 2, our TranSUN achieves the best performance as it is the only method that maintains unbiasedness across all distributions. Subsequently, we provide an analysis of the biasedness of the baselines. The MAE model overestimates the mean on left-skewed and underestimates it on right-skewed data due to its median modeling approach. WLR exhibits an error of $0.2483$ on RS-ZIG, roughly equivalent to $p_+/(1 - p_+) = 0.25$ ($p_+$ is the positive rate of RS-ZIG), consistent with the derivations in [9]. ZILN and OptDist both show significant biasedness on certain distributions, owing to these severely violating the LogNormal or ZILN distribution assumptions. MDME overestimates the mean on left-skew and underestimates it on right-skew data, likely because it estimates the mean of mode buckets under mode sub-distribution, which also explains its biasedness on SM-BU and SM-U as sampling error may result in severe mode estimation error on these data. TPM underestimates the mean on left-skew and overestimates it on right-skew data, likely because it learns the average of quantiles. Note that increasing bin number can reduce its bias, as shown in Sec E.1. CREAD presents a similar bias problem to TPM, whereas slightly alleviated due to its EAD binning method [38].

**Comparison of GTS instance models.** Some common $\mathcal{L}_{\mathcal{H}_q}$ and $\kappa$ are selected to devise different instances of GTS, providing reference results for its application. As exhibited in Table 15, in terms of $\mathcal{L}_{\mathcal{H}_q}$, MSE loss is empirically the most versatile loss as it fastly converges to the unbiased solution in all distributions, whose $\mathbb{Q}$ corresponds to the arithmetic mean point. Among other losses, MAE loss presents better convergence on right-skewed data, while MSPE and MAPE loss both converge more effectively on left-skewed and symmetric data. Regarding function $\kappa$, all candidates exhibit unbiasedness consistent with TranSUN. The performance of $\kappa$ on real-world data is shown in Table 10.

Table 3: Comparison of transformed MSE (T-MSE) and TranSUN under different transformations T on CIKM16 train and DTMart train. Bold indicates that **gain <-70%**. The differences are statistically significant at 0.05 level.

| Methods | T(·) | CIKM16 train | | DTMart train | |
|---|---|---|---|---|---|
| | | TRE ↓ | MRE ↓ | TRE ↓ | MRE ↓ |
| T-MSE | $T(y) = \ln(y+1)$ | 0.3468 | 0.3352 | 0.2894 | 0.1432 |
| **TranSUN** | | **0.0133** (-96.2%) | **0.0171** (-94.9%) | **0.0803** (-72.3%) | 0.0725 (-49.4%) |
| T-MSE | $T(y) = \sqrt{y}$ | 0.1386 | 0.1243 | 0.4388 | 0.3421 |
| **TranSUN** | | **0.0388** (-72.0%) | **0.0283** (-77.2%) | **0.0907** (-79.3%) | **0.066** (-80.7%) |
| T-MSE | $T(y) = \arctan(y)$ | 3.6266 | 10.9861 | 0.6936 | 0.3678 |
| **TranSUN** | | **0.3528** (-90.3%) | **1.2674** (-88.5%) | **0.1554** (-77.6%) | **0.0678** (-81.6%) |

# 5 Experiments on real-world data

## 5.1 Experiment settings

**Datasets and metrics.** We adopt two public datasets (**CIKM16**[1], **DTMart**[2]) from different fields and one exclusive industrial dataset (*abbr.* **Indus**) collected from an e-commerce platform. Details are shown in Appendix D.2.1. Indus aims to predict the Gross Merchandise Value (GMV) of each order paid on the platform, where the Pareto phenomenon occurs, *i.e.* 80% of the GMV being contributed by the top 30% high-priced samples. Hence, we further conduct evaluation solely on the top high-priced samples, *abbr.* **Indus Top** test. Two metrics akin to SRE in Sec. 4.1 are specifically proposed for assessing unbiasedness (see Appendix D.2.2 for detailed discussion), *i.e.* Total Ratio Error (TRE) and Mean Ratio Error (MRE). Specifically, TRE and MRE are defined as $|\sum_{i=1}^{M}(\hat{y}_i - y_i)/\sum_{i=1}^{M}\hat{y}_i|$ and $|1/M \times \sum_{i=1}^{M}(\hat{y}_i - y_i)/\hat{y}_i|$, respectively, where $\hat{y}_i$ is prediction, $y_i$ is ground truth, and $M$ is sample number. We further adopt some metrics commonly used in the literature [43, 21, 23, 38], *i.e.* NRMSE, NMAE, XAUC, and NDCG (NDCG results are shown in Table 11).

**Implementation details.** See Appendix D.2.3 for the detailed introduction. TensorFlow-style implementation of our TranSUN and GTS is exhibited in Code 1. Some optional techniques for optimizing TranSUN's online performance are introduced in Appendix A.4. Moreover, the efficiency analysis of our method is demonstrated in Appendix A.3.

## 5.2 Results

**Effectiveness on eradicating retransformation bias.** Since these target data are right-skewed, commonly-used concave functions, *i.e.* logarithm, square, and arctan transformation, are adopted for performance comparison between transformed MSE and our TranSUN. Because retransformation bias can be viewed as fitting deviation, evaluations are conducted on the training sets of CIKM16 and DTMart. As shown in Table 3, transformed MSE exhibits significant bias issues, with $\arctan$ transformation being the most severe. Differently, TranSUN significantly mitigates the bias (mostly **< -70.0%**) for all transformations. Furthermore, we assessed their performance on different target bins on Indus, where bins are obtained by dividing ascending targets into equal frequencies. In Fig. 3c, LogMSE and **LogSUN** denotes transformed MSE and TranSUN with $T(y) = \ln(y+1)$, respectively, with Signed TRE used as metric. As illustrated, LogSUN substantially alleviated bias across all bins compared to LogMSE. Moreover, LogMSE's bias issue gets more severe when the target value grows larger, and meanwhile, our LogSUN's debiasing intensity gets higher as well. Fig. 3c further shows $\kappa$ gradually decreases as target $y$ increases, to provide relatively stable $\kappa \cdot y$ for smoothing $\mathcal{L}_{\text{sun}}^{\ln}$.

**Comparison with post-hoc correction methods.** To further demonstrate the debiasing capability of our method, our **LogSUN** is compared against two classic bias correction methods, *i.e.* Normal Theory Estimates (NTE) [11] and Smearing [12], alongside a mainstream calibration method, *i.e.* SIR [10], on CIKM16 test. Different from LogSUN, all the baselines are post-hoc methods, which naturally confer them an advantage in the comparison. Nevertheless, as shown in Table 4, our method achieves comparable performance with these baselines. Among them, the poor performance of NTE

---

[1] https://competitions.codalab.org/competitions/11161

[2] https://www.kaggle.com/datasets/datatattle/dt-mart-market-mix-modeling/

Table 4: Comparison with the post-hoc correction methods on CIKM16 test. The **best** are in boldface and the second best are underlined.

| Models | Correction methods | TRE↓ | MRE↓ | NRMSE↓ | NMAE↓ |
|---|---|---|---|---|---|
| LogMSE | - | 0.3451 | 0.3667 | 0.6189 | 0.4528 |
| LogMSE | NTE [11] | 0.1262 | 0.1442 | 0.5718 | 0.4369 |
| LogMSE | Smearing [12] | 0.0175 | 0.0477 | **0.5617** | 0.4335 |
| LogMSE | SIR [10] | 0.0198 | **0.0236** | 0.5821 | **0.4309** |
| **LogSUN** | - | **0.0123** | 0.0439 | 0.5625 | 0.4333 |

Table 5: Comparison of LogSUN's different bias modeling schemes on CIKM16 test, where bold indicates the **best**.

| Scheme ID | $\mathcal{L}_{\text{sun}}^{\text{T}}$ | TRE↓ | MRE↓ | NRMSE↓ | NMAE↓ | XAUC↑ |
|---|---|---|---|---|---|---|
| S0 | $\left(z(x;\theta_z) - \text{stop\_grad}\left[y - \exp(f(x;\theta))\right]\right)^2$ | 0.0689 | 0.0829 | 0.6153 | 0.4459 | 0.6711 |
| S1 | $\left(z(x;\theta_z) - \text{stop\_grad}\left[\exp(f(x;\theta))/y\right]\right)^2$ | 0.1288 | 0.1036 | 0.6255 | 0.4387 | 0.6730 |
| S2 (Ours) | $\left(z(x;\theta_z) - \text{stop\_grad}\left[y \cdot \exp(-f(x;\theta))\right]\right)^2$ | **0.0123** | **0.0439** | **0.5625** | **0.4333** | **0.6740** |

Table 6: Results of extending our method to advanced regression models on CIKM16 test. Except for XAUC, metric gains are statistically significant at 0.05 level.

| Extensions | TRE↓ | MRE↓ | NRMSE↓ | NMAE↓ | XAUC↑ |
|---|---|---|---|---|---|
| WLR [9] + GTS | 0.0286 (-88.72%) | 0.0714 (-78.36%) | 0.5814 (-33.20%) | 0.4735 (-22.42%) | 0.6737 (-0.07%) |
| ZILN [40] + GTS | 0.0250 (-94.31%) | 0.0223 (-95.04%) | 0.5580(-12.81%) | 0.4308 (-7.53%) | 0.6745 (0.01%) |
| MDME [21] + GTS | 0.0828 (-81.38%) | 0.0254 (-73.98%) | 1.0458 (-33.60%) | 0.7874 (-30.85%) | 0.6743 (0.19%) |
| OptDist [41] + GTS | 0.0234 (-93.88%) | 0.0248 (-94.96%) | 0.5604 (-10.84%) | 0.4375 (-4.66%) | 0.6748 (-0.09%) |

Table 7: Comparison results of advanced regression methods on CIKM16 test and DTMart test. The **best** results are in boldface and the second best are underlined.

| Models | CIKM16 test | | | | | DTMart test | | | | |
|---|---|---|---|---|---|---|---|---|---|---|
| | TRE↓ | MRE↓ | NRMSE↓ | NMAE↓ | XAUC↑ | TRE↓ | MRE↓ | NRMSE↓ | NMAE↓ | XAUC↑ |
| MSE | 0.1139 | 0.1687 | 0.6192 | 0.5024 | 0.6693 | 0.1028 | 0.1349 | 0.4893 | 0.2579 | 0.9369 |
| MAE | 0.0819 | 0.1281 | 0.6837 | 0.4285 | 0.6747 | 0.1655 | 0.0917 | 0.4940 | 0.2237 | 0.9388 |
| LogMSE | 0.3451 | 0.3667 | 0.6189 | 0.4528 | 0.6745 | 0.2889 | 0.1448 | 0.5232 | 0.2494 | 0.9440 |
| SqrtMSE | 0.1339 | 0.1509 | 0.5839 | 0.4222 | 0.6753 | 0.4392 | 0.3445 | 0.5621 | 0.3115 | 0.9422 |
| WLR [9] | 0.2536 | 0.3299 | 0.8703 | 0.6103 | 0.6742 | 0.0916 | 0.7488 | 1.0178 | 0.5770 | 0.9249 |
| ZILN [40] | 0.4391 | 0.4498 | 0.6400 | 0.4659 | 0.6744 | 0.1165 | 0.0838 | 0.5107 | 0.2119 | 0.9445 |
| MDME [21] | 0.4448 | 0.0976 | 1.5749 | 1.1387 | 0.6730 | 0.2149 | 0.0861 | 0.9654 | 0.3395 | 0.9296 |
| TPM [23] | 0.0278 | 0.0548 | 0.5632 | 0.4358 | **0.6758** | 0.0742 | 0.0802 | 0.4998 | 0.2064 | 0.9443 |
| CREAD [38] | 0.0232 | 0.0639 | 0.5639 | 0.4274 | 0.6753 | 0.1533 | 0.1574 | 0.4409 | 0.2583 | 0.9443 |
| OptDist [41] | 0.3825 | 0.4920 | 0.6285 | 0.4589 | 0.6754 | 0.1422 | 0.0850 | 0.4418 | 0.2121 | **0.9450** |
| **LogSUN** | **0.0123** | **0.0439** | 0.5625 | 0.4333 | 0.6740 | **0.0731** | 0.0790 | **0.4151** | **0.1994** | 0.9394 |
| **SqrtSUN** | 0.0347 | 0.0545 | **0.5527** | **0.4212** | 0.6752 | 0.0890 | **0.0656** | 0.4194 | 0.2031 | 0.9391 |

is attributed to the transformed target not adhering to the Gaussian distribution assumptions. In contrast, Smearing and SIR exhibit superior results, as they are free from distributional assumptions.

**Performance comparison with state-of-the-arts.** Our TranSUN is compared against common regression loss (*e.g.* MSE, MAE), transformed models (*e.g.* LogMSE, SqrtMSE), advanced classification-based regression models (*e.g.* TPM, MDME), and fancy value regression models (*e.g.* ZILN, OptDist) on CIKM16 test and DTMart test. In Table 7, **LogSUN** and **SqrtSUN** denote our method with $\text{T}(y)$ set to $\ln(y + 1)$ and $\sqrt{y}$, respectively. As illustrated, our method achieves the best performance across our TRE and MRE metrics, as well as the commonly used NRMSE and NMAE metrics. Note that applying TranSUN usually results in a slight decline in XAUC. However,

Table 8: Comparison of advanced regression methods on Indus test, with bold indicating the **best**.

| Models | Indus test | | | | | Indus Top test | | | | |
|---|---|---|---|---|---|---|---|---|---|---|
| | TRE↓ | MRE↓ | NRMSE↓ | NMAE↓ | XAUC↑ | TRE↓ | MRE↓ | NRMSE↓ | NMAE↓ | XAUC↑ |
| MSE | 0.3253 | 1.6568 | 2.4540 | 0.6018 | 0.7737 | 0.7719 | 2.5136 | 1.6605 | 0.5845 | 0.6703 |
| LogMSE | 0.6080 | 0.7463 | 2.4105 | 0.5871 | 0.7819 | 1.1334 | 2.4296 | 1.6130 | 0.5475 | 0.6593 |
| LogMAE | 0.4685 | 0.8314 | 2.4394 | 0.5691 | 0.7816 | 0.8973 | 2.5427 | 1.6427 | 0.5273 | 0.6567 |
| WLR [9] | 0.6940 | 0.7399 | 2.5622 | 0.6419 | 0.7751 | 0.7883 | 1.3097 | 1.6403 | 0.5673 | 0.6608 |
| ZILN [40] | 0.2945 | 0.4159 | 2.6586 | 0.5890 | 0.7809 | 0.7547 | 1.5717 | 1.5543 | 0.5277 | 0.6661 |
| MDME [21] | 0.3987 | 0.4585 | 2.4880 | 0.6225 | 0.7790 | 0.7986 | 1.5056 | 1.5789 | 0.5494 | 0.6677 |
| TPM [23] | 0.0652 | 0.1350 | 2.3981 | 0.6186 | 0.7799 | 0.4218 | 0.9261 | 1.5706 | 0.5052 | 0.6730 |
| CREAD [38] | 0.0917 | 0.1180 | 2.3353 | 0.5815 | 0.7808 | 0.6484 | 1.0733 | 1.5825 | 0.5191 | 0.6693 |
| OptDist [41] | 0.2867 | 0.3650 | 2.3486 | 0.5780 | **0.7822** | 0.7247 | 1.4571 | 1.5614 | 0.5212 | 0.6679 |
| **LogSUN** | **0.0197** | **0.0410** | **2.2765** | **0.5683** | 0.7815 | **0.3730** | **0.8602** | **1.4969** | **0.4801** | **0.6749** |

Table 9: Results of the online A/B test, with the 95% confidence interval (CI-95%) and $p$-value.

| Scenarios | Product recommendation | | | | Short video recommendation | |
|---|---|---|---|---|---|---|
| A/B tests
Traffics & periods | **LogSUN (Ours)** *vs.* LogMAE (Base)
15% *vs.* 20% for 1 week | | **LogSUN (Ours)** *vs.* LogMSE
15% *vs.* 9% for 1 week | | **LogSUN (Ours)** *vs.* LogMAE (Base)
6% *vs.* 6% for 1 month | |
| Metrics | GMV | GMVPI | GMV | GMVPI | GMV | GMVPI |
| Gains
95%CI
$p$-value | **+1.06%**
[ 0.52% , 1.59% ]
0.0001 | **+0.92%**
[ 0.32% , 1.51% ]
0.0024 | **+1.02%**
[ 0.51% , 1.54% ]
0.0001 | **+0.75%**
[ 0.18% , 1.32% ]
0.0096 | **+1.83%**
[ 0.50% , 3.17% ]
0.0065 | **+1.57%**
[ 0.24% , 2.89% ]
0.0198 |

this trade-off remains favorable as discussed in Appendix A.1. Furthermore, as shown in Table 8, consistent superiority of our LogSUN is observed on Indus test, where its XAUC for top high-priced samples also achieves the best performance (**> +0.015** uplift to LogMSE).

**Extension to state-of-the-arts.** To validate the effectiveness of GTS as a debiasing framework, we apply our method to several state-of-the-art models with significant bias issues on CIKM16 test, where $\kappa(f(x;\theta_q)) = 1/(f(x;\theta_q)+1)$ is adopted in Eq. (9). As depicted in Table 6, the bias issues of classification-based models (*i.e.* MDME, WLR) and value regression models (*i.e.* ZILN, OptDist) are all significantly mitigated (**< -70.0%** in TRE and MRE) by integrating our approach.

**Online A/B test.** In Taobao App (DAU > 300M), our **LogSUN** is deployed to serve as a pointwise ranking model for GMV target in two recommendation scenarios (*i.e.* product and short video recommendation in *Guess What You Like* business domain in Taobao App homepage), respectively, to optimize object distribution (see Appendix E.2 for details), which is compared against the previous online base (LogMAE) and LogMSE. Two common metrics, GMV and GMVPI (GMV Per Impression), are adopted to assess online performance. Due to the large variability of these GMV-related metrics, we allocated either substantial traffic (*e.g.* 15% of the total traffic for LogSUN) or a sufficiently long period (*e.g.* one month) for A/B tests to obtain statistically significant results. As shown in Table 9, LogSUN achieves significant gains (*e.g.* **> +1.0%** GMV gains) over all the baselines. Now, LogSUN has been deployed to serve the major traffic.

### 5.3 Ablation studies

**Comparison of different GTS instance models.** To offer reference results for GTS's application on real-world data, some common $\mathcal{L}_{\mathcal{H}_q}$ and $\kappa$ are selected to build different instances of GTS for comparison. As shown in Table 10, MSE loss is the best-performing $\mathcal{L}_{\mathcal{H}_q}$, while MAE loss also performs well as the target data are right-skewed, which is consistent with the results in Table 15. Regarding function $\kappa$, compared to the one used in TranSUN, other $\kappa$ candidates all bring about poorer loss convergence, which is probably because the variances of their $\kappa \cdot y$ are all too large (shown in Table 17), leading to an excessively steep optimization landscape for $\theta_z$.

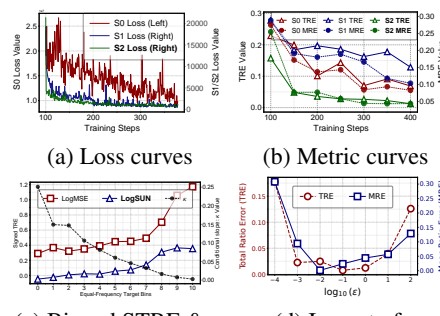

(a) Loss curves    (b) Metric curves

(c) Binned STRE & $\kappa$    (d) Impact of $\epsilon$

Figure 3: Visualization results across experiments. (a) Training loss curves for LogSUN with different schemes on CIKM16. (b) Training curves of TRE and MRE for LogSUN with different schemes on CIKM16. (c) Signed TRE (STRE) and average slope $\bar{\kappa}$ on divided target bins of Indus test. (d) The sensitivity of LogSUN to $\epsilon$ on CIKM16 test.

**Comparison of different bias modeling schemes.** To show the superiority of the multiplicative bias modeling scheme in Eq. (4), two other intuitive schemes are compared with ours on the LogSUN model, as presented in Table 5, where notably neither of the scheme candidates (S0, S1) belongs to GTS family. The impact of different schemes on model training is illustrated in Fig. 3a, 3b. As visible, despite theoretically unbiased, S0 scheme exhibits poor convergence performance, likely due to the high loss variance caused by its additive modeling scheme. S1 scheme converges well but suffers from significant bias, as it is not theoretically unbiased, as discussed in Sec. 3.2. In contrast, our scheme (S2) achieves both superior convergence and theoretical unbiasedness.

Table 10: Comparison of different GTS instances ($\mathrm{T}(y) = \ln(y+1)$) on CIKM16 test . The **best** results are in boldface and the second best are underlined.

| $\mathcal{L}_{\mathcal{H}_q}$ | $\kappa(\cdot)$ | TRE $\downarrow$ | MRE $\downarrow$ | NRMSE $\downarrow$ | NMAE $\downarrow$ | XAUC $\uparrow$ |
|---|---|---|---|---|---|---|
| MSE | $\kappa(u) = 1/(\|\mathrm{T}^{-1}(u)\| + \epsilon)$ | **0.0123** | 0.0439 | **0.5625** | 0.4333 | **0.6740** |
| MAE | $\kappa(u) = 1/(\|\mathrm{T}^{-1}(u)\| + \epsilon)$ | 0.0145 | **0.0370** | 0.5627 | 0.4336 | **0.6725** |
| MSPE | $\kappa(u) = 1/(\|\mathrm{T}^{-1}(u)\| + \epsilon)$ | 0.1077 | 0.0679 | 0.5856 | 0.4595 | 0.6731 |
| MAPE | $\kappa(u) = 1/(\|\mathrm{T}^{-1}(u)\| + \epsilon)$ | 0.0340 | 0.0692 | 0.5630 | **0.4316** | 0.6734 |
| MSE | $\kappa(u) = 1/(\|u\| + \epsilon)$ | 0.0399 | 0.0680 | 0.5898 | 0.4753 | 0.6708 |
| MSE | $\kappa(u) = \|u\|$ | 0.1810 | 0.2483 | 0.6328 | 0.5423 | 0.6633 |

**Architectural sensitivity.** Since our method introduces an extra bias learning branch to transformed MSE, we explore the impact of different parameter sharing schemes between the added branch and the original model, which is conducted on the LogSUN model. In Table 16, sharing parameters from bottom layers (*e.g.* embedding layers) has minimal impact on performance, whereas further sharing more parameters within MLP results in a noticeable performance degradation, which might be inferred that the added task primarily affects the downstream MLP-modeled fitting function rather than the distribution of latent embedding.

**Impact of hyperparameters.** We study the impact of the only hyperparameter $\epsilon$ on the Log-SUN model using CIKM16. As visible in Fig. 3d, LogSUN stably demonstrates superior performance when $\epsilon$ is between $0.01$ and $1.0$. As $\epsilon$ further approaches zero, the performance rapidly degrades. This may be because $\mathrm{T}^{-1}(f(x;\theta))$ tends to be near zero at the early stages of training, thus $y/(\|\mathrm{T}^{-1}(f(x;\theta))\| + \epsilon)$ gets extremely large when $\epsilon$ is close to zero as well, causing significant loss oscillation and hindering model convergence. Conversely, performance also degrades when $\epsilon \geq 100$, likely due to $y/(\|\mathrm{T}^{-1}(f(x;\theta))\| + \epsilon)$ being too small to be learned within the fitting scope of $z(x;\theta_z)$, thus leading to the model underfitting.

# 6 Conclusion

In this paper, we propose a novel preemptive paradigm to eradicate retransformation bias intrinsically from the regression models to challenge the conventional ones, as the first work in the recommender system field to address the long-neglected bias problem. Specifically, a novel method, TranSUN, is proposed with a joint bias learning manner to offer theoretically guaranteed unbiasedness under empirical superior convergence. It is further generalized into a novel generic regression model family, Generalized TranSUN (GTS), which not only offers theoretical insights into TranSUN but also serves as a generic framework for flexibly developing various bias-free models. Extensive experiments demonstrate the superiority and broad applicability of our methods.

**Limitations and future works.** Regarding real-world applications, although the effectiveness of TranSUN has been validated across data from various domains, these target distributions are all right-skewed. Hence, we expect more exploration for its application where the target distributions are not right-skewed. Moreover, a limitation of our method is the decline in XAUC, which conforms to the No Free Lunch (as discussed in Appendix A.1), thus integrating isotonicity into TranSUN is a challenging but interesting direction. In the future, we plan to investigate GTS further, *e.g.* exploring a general methodology to identify the optimal point $\mathbb{Q}$ and function $\kappa$ on any given data.

# Acknowledgments

We deeply appreciate Han Wu, Chi Li, Kun Yuan, Runfeng Zhang, Weiqiu Wang, Longbin Li, Xin Yao, Nan Hu, Chuan Wang, and Hanqi Jin for their insightful suggestions and discussions. We thank Kewei Zhu, Yuxin Duan, Hao Fang, and Gaofeng Li for necessary supports about online serving. We thank anonymous reviewers for their constructive feedback and helpful comments.

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

# A   Frequently Asked Questions (FAQ)

## A.1   Root reason for the trade-off between point accuracy (e.g. NRMSE) and XAUC

Here we elaborate on the root cause of the trade-off phenomenon between point accuracy and XAUC, and further highlight the limitations of XAUC. This limitation leads to the *inconsistency* between the offline XAUC metric and online business metrics (as shown in Table 9), making XAUC no longer suitable for offline assessment. To address this, we supplemented the offline evaluation with two more consistent offline metrics, **high-value XAUC** (as shown in Table 8) and a commonly used ranking metric **NDCG** (as shown in Table 11), on both of which our model demonstrates superior performance. For clarity, let us start with a concrete case as follows.

**An illustrative case for trade-off:**   We first sample two inputs $\{x_1, x_2\} \sim \mathcal{P}(X)$, and then sample 3 groups of target $y$ from $\mathcal{P}(Y|x_1)$ and $\mathcal{P}(Y|x_2)$, respectively, and let the samples be $\{3, 3, 3\} \sim \mathcal{P}(Y|x_1)$ and $\{8, 1, 1\} \sim \mathcal{P}(Y|x_2)$, and let $T(x) = \log(x)$. Then, **1)** The LogMSE prediction is $f(x_1) = T^{-1}(\mathbb{E}[T(Y)|x_1]) = \exp((\log(3) + \log(3) + \log(3))/3) = 3$, $f(x_2) = T^{-1}(\mathbb{E}[T(Y)|x_2]) = \exp((\log(8) + \log(1) + \log(1))/3) = 2$, where $f(x_1) > f(x_2)$ and the bias occurs on the sample $X = x_2$; **2)** The TranSUN/GTS prediction is $f(x_1) = \mathbb{E}[Y|x_1] = (3 + 3 + 3)/3 = 3$, $f(x_2) = \mathbb{E}[Y|x_2] = (1 + 1 + 8)/3 = 10/3$, where $f(x_1) < f(x_2)$; **3)** The descending ranking result of LogMSE is $\{3, 3, 3, 8, 1, 1\}$, with XAUC = 0.333, NRMSE = 0.324, and TRE = 4/15, while the result of TranSUN/GTS is $\{8, 1, 1, 3, 3, 3\}$, with XAUC = 0.233, NRMSE = 0.301, and TRE = 0; **4)** It is observed that a trade-off occurs between point accuracy (NRMSE, TRE) and XAUC, *i.e.* TranSUN/GTS achieves better point accuracy (NRMSE, TRE) while XAUC worsens.

**Root cause of the trade-off:**   **1)** *Reason for the improvement of high-value XAUC & overall point accuracy metric*: TranSUN/GTS compensates underestimation brought by retransformation bias, resulting in better point and rank accuracy of high-value samples (*e.g.* for sample $(x_2, 8)$, relative error improves from 0.75 to 0.58, and its rank order moves from 4th to 1st, consistent with results on Indus Top (Table 8)), which significantly improves the overall point accuracy (*e.g.* improvements in NRMSE & TRE in the above case) since these metrics are **value-aware** (*i.e.* metric calculation aware of absolute value of sample label); **2)** *Reason for the drop of overall XAUC*: Correcting retransformation bias also leads to some hard low-value samples being overestimated (*e.g.* predictions for the two samples $(x_2, 1)$ increase from 2 to 10/3, consistent with Fig. 3c), which dominates the overall XAUC drop since there are far more hard low-value samples than high-value ones in highly right-skewed data and XAUC metric is **value-agnostic** (*i.e.* metric calculation only aware of the relative rank order of sample label but not aware to its absolute value).

**Overall XAUC is inconsistent with online metrics, thus we use high-value XAUC and NDCG:**
**1)** The core goal of a recommender system is to *improve online business metrics*, while XAUC is merely an intermediate metric. For business target modeled by regression (GMV, watch time, *etc.*), the event of ranking high-valued samples in the correct position is far more important for online metrics than that of low-valued samples. However, these two events are equally weighted (i.e. value-agnostic) in the calculation of XAUC, thereby being dominated by abundant low-valued samples when data are highly right-skewed, which probably contribute only a small portion of online metrics (*e.g.* 70% of the top low-valued samples contribute only 20% of GMV in Indus). Therefore, overall XAUC is no longer consistent with online metrics; **2)** To address this, we supplemented the offline assessment with a more consistent metric, high-value XAUC (Table 8), based on which we achieved improved online results (Table 9). Additionally, NDCG also addresses XAUC's limitation, so we also provide their results. As shown in Table 11, NDCG is computed within all samples (NDCG@All) and the top 10% high-value ones (NDCG@10%), respectively. Key observations include: *a)* TranSUN performs better in NDCG@All than in XAUC since NDCG is more robust to low-value samples. *b)* TranSUN shows superior performance in NDCG@10%, consistent with the reasons described in the 2nd point.

## A.2   Convergence analysis for $\mathcal{L}_{\text{sun}}^{\text{T}}$, $\mathcal{L}_{\text{MSE}}^{\text{T}}$, and $\mathcal{L}_{\text{MSE}}$

As for convergence rate bound, since their loss formulations are all squared errors, the theoretical convergence rate bound using SGD is $O(\frac{1}{\sqrt{K}})$ under the Lipschitz assumption, where $K$ is update iteration number, according to [6]. The key difference in the convergence performance of $\mathcal{L}_{\text{sun}}^{\text{T}}$, $\mathcal{L}_{\text{MSE}}^{\text{T}}$,

Table 11: NDCG metric results of our method and advanced baselines on CIKM16 test and DTMart test. The **best** results are in boldface.

| Model | CIKM16 test | | DTMart test | |
|---|---|---|---|---|
| | NDCG@All | NDCG@10% | NDCG@All | NDCG@10% |
| MSE | 0.9180 | 0.5201 | 0.9770 | 0.9692 |
| MAE | 0.9497 | 0.5654 | 0.9867 | 0.9665 |
| LogMSE | 0.9498 | 0.5517 | 0.9960 | 0.9736 |
| WLR [9] | 0.9475 | 0.5433 | 0.9693 | 0.9609 |
| ZILN [40] | 0.9485 | 0.5781 | 0.9963 | 0.9756 |
| MDME [21] | 0.9391 | 0.5671 | 0.9763 | 0.9639 |
| TPM [23] | 0.9502 | 0.5888 | 0.9960 | 0.9772 |
| CREAD [38] | **0.9504** | 0.5881 | 0.9962 | 0.9774 |
| OptDist [41] | 0.9500 | 0.5852 | 0.9965 | 0.9770 |
| **LogSUN** | 0.9499 | **0.5904** | **0.9967** | **0.9779** |

and $\mathcal{L}_{\text{MSE}}$ lies in their *smoothness*. This is because the variance and the outlier proportion of the target are different, resulting in optimization landscapes with different degrees of smoothness. As shown in Table 12, we present the target variance (**Var**), the variance divided by squared expectation (**V/E2**), and the outlier proportion (**Outlier%**) outside of 3 times the standard deviation of the three losses, where $\text{T}(x) = \log(x)$. As shown, $\mathcal{L}_{\text{sun}}^{\text{T}}$ has a smaller variance and a better outlier distribution than the vanilla MSE, thus $\mathcal{L}_{\text{sun}}^{\text{T}}$ converges more smoothly (Fig.1). Therefore, TranSUN/GTS essentially *decomposes a difficult task into two easier and smoother tasks.*

Table 12: Convergence analysis for $\mathcal{L}_{\text{sun}}^{\text{T}}$, $\mathcal{L}_{\text{MSE}}^{\text{T}}$, and $\mathcal{L}_{\text{MSE}}$

| Losses | CIKM16 | | | DTMart | | |
|---|---|---|---|---|---|---|
| | Var | V/E2 | Outlier% | Var | V/E2 | Outlier% |
| $\mathcal{L}_{\text{MSE}}$ | 16440 | 0.41 | 1.03% | 97494 | 12.1 | 0.78% |
| $\mathcal{L}_{\text{sun}}^{\text{ln}}$ | 0.63 | 0.36 | 0.59% | 2.327 | 0.93 | 0.32% |
| $\mathcal{L}_{\text{MSE}}^{\text{ln}}$ | 0.62 | 0.02 | 0.29% | 1.87 | 0.15 | 0.28% |

## A.3 Efficiency analysis for the proposed TranSUN

As different parameter sharing architectures of TranSUN are explored in Appendix E.4, here we supplement the results (Table 16) with TranSUN's additional memory (Mem) and computational overhead (MFLOPS), and calculate the relative increase (OH) compared with T-MSE. As shown in Table 13, the only-sharing-embedding architecture scheme is the most cost-effective, maintaining optimal performance while introducing much little overhead. On the other hand, we provide additional information on the resource overhead associated with the online TranSUN model in the e-commerce platform's recommendation system, which achieves a *high ROI* measure (with Table 9): **1) Training:** Under the same resources (800 workers), the update iteration (global step) number per second of LogMSE and TranSUN were 89.2 and 86.8, respectively. Therefore, applying TranSUN only reduces the training speed by 2.7%, with the training time increased by 2.78%; **2) Inference:** The mean and p99 latency of the LogMSE were 27.8ms and 49.8ms, respectively, while the mean and p99 latency of TranSUN were 28.1ms and 51.6ms, respectively; **3) Deployment:** In the online service with 300 million daily active users, deploying TranSUN did not increase CPU resources and only increased GPU resources by 5%.

## A.4 Optional techniques for optimizing LogSUN's online performance

As mentioned in Appendix A.1, TranSUN likely results in a certain decrease in the overall XAUC, particularly for low-priced orders. Although these orders only contribute 20% of the total GMV, they represent the majority in view of the total number of transaction orders. Therefore, the reduced

Table 13: Comparison results of the LogSUN model under different parameter sharing architectures on CIKM16 test. "$X/Y$" denotes sharing the first $X$-th layers of the network with totally $Y$ layers. The results with the **best ROI** are in boldface.

| Parameter sharing schemes | Mem (MB) | OH (%) | MFLOPS | OH (%) |
|---|---|---|---|---|
| 0/1 Embedding + 0/3 MLP | 610.46 | 100.00 | 350.86 | 100.00 |
| **1/1 Embedding + 0/3 MLP** | **305.29** | **0.02** | **190.52** | **8.60** |
| 1/1 Embedding + 1/3 MLP | 305.27 | 0.01 | 186.12 | 6.09 |
| 1/1 Embedding + 2/3 MLP | 305.24 | 0.00 | 177.62 | 1.25 |
| 1/1 Embedding + 3/3 MLP | 305.23 | 0.00 | 175.47 | 0.02 |

ranking accuracy of these orders' GMV estimates deteriorates the accuracy of their overall ranking score, which in turn deteriorates the online metrics regarding the number of transaction orders. Therefore, it is crucial to lift the ranking accuracy of these samples under TranSUN's modeling. Here are two techniques that have been validated as effective in our scenarios: **1) Selecting a suitable combination of ranking losses:** As shown in lines 77-79 of the Code 1, selecting these two ranking losses carefully has been shown to improve the overall XAUC by > +0.01 on our private industrial data. **2) Fusion with the original predictions:** As shown in lines 51-52 of the Code 1, a pre-defined fusion threshold can be used to restrict TranSUN's functioning scope to high-priced samples, which improved the overall XAUC by > +0.003 on our private data. To select the threshold, we first divided the sample into 50 bins with equal frequency based on their original prediction values in ascending order. Then, we calculated the XAUC results of the original and debiased prediction values within each bin, respectively. We found that as the bin number increased, the XAUC gain of debiasing gradually shifted from negative to positive. Therefore, the left boundary of the bin that first achieved a positive XAUC gain was adopted as the fusion threshold. Once this threshold is determined, it does not need to be adjusted anymore, unless the data distribution shifts significantly.

### A.5 Comparing GTS with probabilistic modeling methods (PMM), *e.g.* Mixture Density Networks (MDN), Normalizing Flows (NF)

A comprehensive comparison is shown in Fig. 4. Specifically, when our GTS serves as a point estimator, it has three advantages over PMM: **1) Flexibility:** It can be flexibly plugged into any model with minor revision (as validated in Table 6). Another advantage is that in industrial practice TranSUN can be fully warmed-up from the online base model, further accelerating TranSUN's convergence; **2) Lightweight:** It requires much less resource than PMM. For example, MDN's overhead grows linearly with mixture components, and NF incurs significant overhead due to Jacobian computations; **3) Training stability:** PMM is much harder to converge than GTS given their complex assumptions. For example, on Indus, OptDist [41] (an MDN case) needs 2.13x the iterations TranSUN requires to reach batch-NRMSE=3.0.

In addition, with minor revision to GTS's model assumption (*i.e.* modeling constant $\sigma$ in Eq. (11) as a function of $x$), GTS can also directly model the conditional probability $\mathcal{P}(Y|x)$ in the CTM manner [16]. Specifically, GTS's assumption formulation turns to

$$-z(x;\theta_z) + Y_x \cdot s(x;\theta_s) \cdot \kappa\big(\mathbb{Q}_{y\sim\mathcal{P}(Y|x)}\big[\mathrm{T}(y)\big]\big) \sim \mathcal{N}(0,1), \tag{12}$$

where $s(x;\theta_s) > 0$ actually models the reciprocal of the standard deviation $\sigma(x)$. In training stage, according to [29, 16], one can establish the scoring-rule-based loss $\mathcal{L}_{sr}(x,y;\mathbf{h})$ as

$$\mathcal{L}_{sr}(x,y;\mathbf{h}) = -\mathbb{E}_{(x,y)\sim\mathcal{D}}\Big[\sum_{i=1}^{N} \mathbf{1}\{y \le v_i\}\log(\Phi(\mathbf{h}(x,v_i))) + \mathbf{1}\{y > v_i\}\log(1 - \Phi(\mathbf{h}(x,v_i)))\Big], \tag{13}$$

where the $\mathbf{1}\{\cdot\}$ is the indicator function, $\{v_i\}_{i=1}^{N}$ denotes pre-defined ascending grid points of $y$, $\Phi(u) = \frac{1}{2}[1 + \mathrm{erf}(\frac{u}{\sqrt{2}})]$, and $\mathbf{h}(x,y)$ is defined as

$$\mathbf{h}(x,y) = -z(x;\theta_z) + y \cdot s(x;\theta_s) \cdot \mathrm{stop\_grad}\big[\kappa\big(f(x;\theta_q)\big)\big]. \tag{14}$$

Therefore, the total loss is formulated as

$$\mathcal{L}_{\mathrm{GTS}}^{\mathrm{pmm}} = \mathcal{L}_{\mathcal{H}_q}\big(x,\mathrm{T}(y);\theta_q\big) + \mathcal{L}_{sr}(x,y;\mathbf{h}). \tag{15}$$

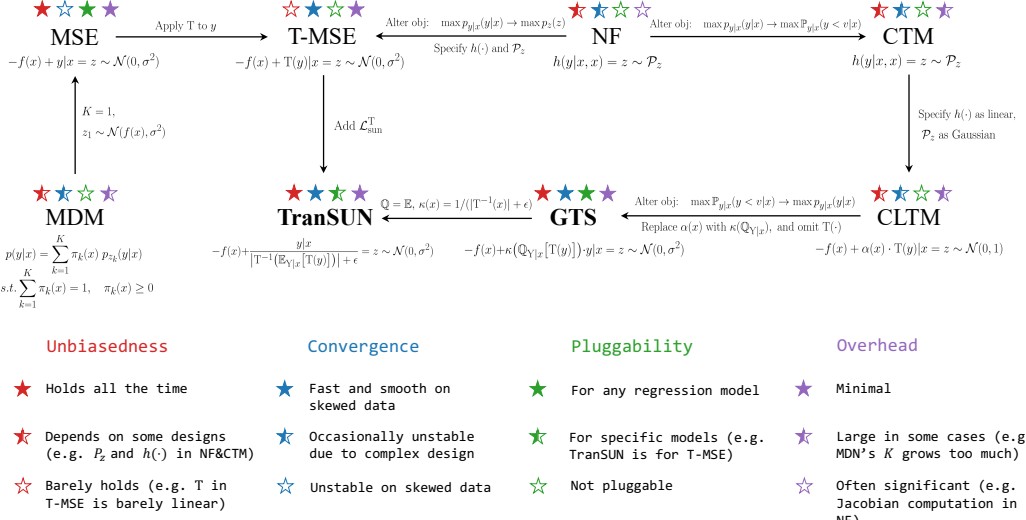

Figure 4: A graphical comparison of the advantages and model assumptions among Mean Squared Error (MSE), transformed MSE (T-MSE), Normalizing Flow (NF), Conditional Transformation Models (CTM), Conditional Linear Transformation Models (CLTM), Mixture Density Model (MDM), TranSUN, and Generalized TranSUN (GTS).

In inference stage, our GTS models the conditional probability with

$$\mathcal{P}(Y < v | x) = \Phi(\mathbf{h}(x, v)). \tag{16}$$

### A.6 Principled guidelines for utilizing GTS

The principled guideline is to align the model assumption (Eq.(11)) as closely as possible with the real data, though this requires manual trial-and-error. Our experience is to first fit the data solely using a mainstream regression model as $\mathbb{Q}$, then analyze the variance, skewness, kurtosis, and outlier distribution of $Y * \kappa(\mathbb{Q})$ to select a suitable $\kappa$ form, which determines the final GTS model. This significantly reduces training resources when tuning $\kappa$. A more general methodology is left for future work, as stated in the conclusion section.

## B Mathematical Details in Our Approach

### B.1 Derivation of transformed MSE's estimation

The loss function of transformed MSE is exhibited in Eq. (2). Minimizing the loss can result in that

$$
\begin{aligned}
&\min_{\theta} \mathbb{E}_{(x,y)\sim\mathcal{D}}\Big[\big(f(x;\theta) - \mathrm{T}(y)\big)^2\Big] \\
&= \min_{\theta} \int_{(x,y)\sim\mathcal{P}(\mathrm{X},\mathrm{Y})} \big(f(x;\theta) - \mathrm{T}(y)\big)^2 p(x,y)\mathrm{d}x\mathrm{d}y \\
&= \min_{\theta} \int_{x\sim\mathcal{P}(\mathrm{X})} \Big(\int_{y\sim\mathcal{P}(\mathrm{Y}|x)} \big(f(x;\theta) - \mathrm{T}(y)\big)^2 p(y|x)\mathrm{d}y\Big) p(x)\mathrm{d}x \\
&= \min_{\theta} \mathbb{E}_{x\sim\mathcal{P}(\mathrm{X})}\Big[\mathbb{E}_{y\sim\mathcal{P}(\mathrm{Y}|x)}\big[(f(x;\theta) - \mathrm{T}(y))^2\big]\Big] \\
&\Leftrightarrow \forall x \sim \mathcal{P}(\mathrm{X}), \ \min_{f(x;\theta)} \mathbb{E}_{y\sim\mathcal{P}(\mathrm{Y}|x)}\big[(f(x;\theta) - \mathrm{T}(y))^2\big] \\
&\Leftrightarrow \forall x \sim \mathcal{P}(\mathrm{X}), \ \frac{\partial}{\partial f(x;\theta)}\mathbb{E}_{y\sim\mathcal{P}(\mathrm{Y}|x)}\big[(f(x;\theta) - \mathrm{T}(y))^2\big] = 0 \\
&\Leftrightarrow \forall x \sim \mathcal{P}(\mathrm{X}), \ \mathbb{E}_{y\sim\mathcal{P}(\mathrm{Y}|x)}\big[2 \times (f(x;\theta) - \mathrm{T}(y))\big] = 0 \\
&\Leftrightarrow \forall x \sim \mathcal{P}(\mathrm{X}), \ f(x;\theta) = \mathbb{E}_{y\sim\mathcal{P}(\mathrm{Y}|x)}\big[\mathrm{T}(y)\big].
\end{aligned}
\tag{17}
$$

Therefore, $f(x;\theta)$ is actually an estimate of $\mathbb{E}_{y\sim\mathcal{P}(Y|x)}\big[T(y)\big]$.

## B.2 Detailed derivation for TranSUN's estimation

A brief derivation for TranSUN's estimate is exhibited in Eq. (7, 8). Following Eq. (17), we present detailed derivation for TranSUN's estimate $\hat{y}|x$ as

$$
\begin{aligned}
&\min_{\theta_z} \mathcal{L}_{\text{sun}}^{\text{T}} \\
&= \min_{\theta_z} \mathbb{E}_{(x,y)\sim\mathcal{D}}\Big[\Big(z(x;\theta_z) - \text{stop\_grad}\Big[\frac{y}{|T^{-1}\big(f(x;\theta)\big)|+\epsilon}\Big]\Big)^2\Big] \\
&= \min_{\theta_z} \mathbb{E}_{x\sim\mathcal{P}(X)}\Big[\mathbb{E}_{y\sim\mathcal{P}(Y|x)}\big[\big(z(x;\theta_z) - \text{stop\_grad}\big[\frac{y}{|T^{-1}\big(f(x;\theta)\big)|+\epsilon}\big]\big)^2\big]\Big] \\
&\Leftrightarrow \forall x\sim\mathcal{P}(X),\ \min_{z(x;\theta_z)} \mathbb{E}_{y\sim\mathcal{P}(Y|x)}\big[\big(z(x;\theta_z) - \text{stop\_grad}\big[\frac{y}{|T^{-1}\big(f(x;\theta)\big)|+\epsilon}\big]\big)^2\big] \\
&\Leftrightarrow \forall x\sim\mathcal{P}(X),\ \frac{\partial}{\partial z(x;\theta_z)}\mathbb{E}_{y\sim\mathcal{P}(Y|x)}\big[\big(z(x;\theta_z) - \text{stop\_grad}\big[\frac{y}{|T^{-1}\big(f(x;\theta)\big)|+\epsilon}\big]\big)^2\big] = 0 \\
&\Leftrightarrow \forall x\sim\mathcal{P}(X),\ \mathbb{E}_{y\sim\mathcal{P}(Y|x)}\big[2\times\big(z(x;\theta_z) - \frac{y}{|T^{-1}\big(f(x;\theta)\big)|+\epsilon}\big)\big] = 0 \\
&\Leftrightarrow \forall x\sim\mathcal{P}(X),\ z(x;\theta_z) = \mathbb{E}_{y\sim\mathcal{P}(Y|x)}\Big[\frac{y}{|T^{-1}\big(f(x;\theta)\big)|+\epsilon}\Big] = \frac{\mathbb{E}_{y\sim\mathcal{P}(Y|x)}[y]}{|T^{-1}\big(f(x;\theta)\big)|+\epsilon} \\
&\Leftrightarrow \forall x\sim\mathcal{P}(X),\ \hat{y}|x = z(x;\theta_z)\cdot\big(|T^{-1}\big(f(x;\theta)\big)|+\epsilon\big) = \mathbb{E}_{y\sim\mathcal{P}(Y|x)}[y].
\end{aligned}
\tag{18}
$$

## B.3 Detailed derivation for estimation from the model with S1 scheme

As mentioned in Sec. 3.2 and Table 4, the S1 bias modeling scheme formulates the bias learning loss as

$$
\mathcal{L}_{\text{sun}}^{\text{T}}(\text{S1}) = \mathbb{E}_{(x,y)\sim\mathcal{D}}\Big[\Big(z(x;\theta_z) - \text{stop\_grad}\Big[\frac{T^{-1}\big(f(x;\theta)\big)}{y}\Big]\Big)^2\Big].
\tag{19}
$$

Following Eq. (18), we present detailed derivation for its estimate $\hat{y}|x$ as

$$
\begin{aligned}
&\min_{\theta_z} \mathcal{L}_{\text{sun}}^{\text{T}}(\text{S1}) \\
&= \min_{\theta_z} \mathbb{E}_{(x,y)\sim\mathcal{D}}\Big[\Big(z(x;\theta_z) - \text{stop\_grad}\Big[\frac{T^{-1}\big(f(x;\theta)\big)}{y}\Big]\Big)^2\Big] \\
&= \min_{\theta_z} \mathbb{E}_{x\sim\mathcal{P}(X)}\Big[\mathbb{E}_{y\sim\mathcal{P}(Y|x)}\big[\big(z(x;\theta_z) - \text{stop\_grad}\big[\frac{T^{-1}\big(f(x;\theta)\big)}{y}\big]\big)^2\big]\Big] \\
&\Leftrightarrow \forall x\sim\mathcal{P}(X),\ \min_{z(x;\theta_z)} \mathbb{E}_{y\sim\mathcal{P}(Y|x)}\big[\big(z(x;\theta_z) - \text{stop\_grad}\big[\frac{T^{-1}\big(f(x;\theta)\big)}{y}\big]\big)^2\big] \\
&\Leftrightarrow \forall x\sim\mathcal{P}(X),\ \frac{\partial}{\partial z(x;\theta_z)}\mathbb{E}_{y\sim\mathcal{P}(Y|x)}\big[\big(z(x;\theta_z) - \text{stop\_grad}\big[\frac{T^{-1}\big(f(x;\theta)\big)}{y}\big]\big)^2\big] = 0 \\
&\Leftrightarrow \forall x\sim\mathcal{P}(X),\ \mathbb{E}_{y\sim\mathcal{P}(Y|x)}\big[2\times\big(z(x;\theta_z) - \frac{T^{-1}\big(f(x;\theta)\big)}{y}\big)\big] = 0 \\
&\Leftrightarrow \forall x\sim\mathcal{P}(X),\ z(x;\theta_z) = \mathbb{E}_{y\sim\mathcal{P}(Y|x)}\Big[\frac{T^{-1}\big(f(x;\theta)\big)}{y}\Big] = T^{-1}\big(f(x;\theta)\big)\cdot\mathbb{E}_{y\sim\mathcal{P}(Y|x)}\big[\frac{1}{y}\big] \\
&\Leftrightarrow \forall x\sim\mathcal{P}(X),\ \hat{y}|x = \frac{T^{-1}\big(f(x;\theta)\big)}{z(x;\theta_z)} = \frac{1}{\mathbb{E}_{y\sim\mathcal{P}(Y|x)}[\frac{1}{y}]}.
\end{aligned}
\tag{20}
$$

As shown above, the model prediction $\hat{y}|x$ is actually an estimate of $1/\mathbb{E}_{y\sim\mathcal{P}(Y|x)}[\frac{1}{y}]$, which is biased.

### B.4 Detailed derivation for GTS's estimation

Owing to the stop_grad operation, the optimization of the total loss $\mathcal{L}_{\text{GTS}}$ can be decomposed into two independent parts, as

$$\min_{\theta_q, \theta_z} \mathcal{L}_{\text{GTS}} \Leftrightarrow \min_{\theta_q} \mathcal{L}_{\mathcal{H}_q} + \min_{\theta_z} \mathcal{L}_{\text{LTM}}^{\kappa}, \tag{21}$$

where $\mathcal{L}_{\mathcal{H}_q}$ is the conditional point loss and $\mathcal{L}_{\text{LTM}}^{\kappa}$ is the linear transformation loss. The first term $\min_{\theta_q} \mathcal{L}_{\mathcal{H}_q}$ learns a conditional point according to Eq. (10), which is independent of $z(x; \theta_z)$. The second term derives the GTS's estimate $\hat{y}|x$ via

$$
\begin{aligned}
&\min_{\theta_z} \mathcal{L}_{\text{LTM}}^{\kappa} \\
&= \min_{\theta_z} \mathbb{E}_{(x,y)\sim\mathcal{D}} \Big[ \big( z(x;\theta_z) - y \cdot \text{stop\_grad}\big[\kappa\big(f(x;\theta_q)\big)\big] \big)^2 \Big] \\
&= \min_{\theta_z} \mathbb{E}_{x\sim\mathcal{P}(\text{X})} \Big[ \mathbb{E}_{y\sim\mathcal{P}(\text{Y}|x)} \big[ \big( z(x;\theta_z) - y \cdot \text{stop\_grad}\big[\kappa\big(f(x;\theta_q)\big)\big] \big)^2 \big] \Big] \\
&\Leftrightarrow \forall x \sim \mathcal{P}(\text{X}), \ \min_{z(x;\theta_z)} \mathbb{E}_{y\sim\mathcal{P}(\text{Y}|x)} \big[ \big( z(x;\theta_z) - y \cdot \text{stop\_grad}\big[\kappa\big(f(x;\theta_q)\big)\big] \big)^2 \big] \\
&\Leftrightarrow \forall x \sim \mathcal{P}(\text{X}), \ \frac{\partial}{\partial z(x;\theta_z)} \mathbb{E}_{y\sim\mathcal{P}(\text{Y}|x)} \big[ \big( z(x;\theta_z) - y \cdot \text{stop\_grad}\big[\kappa\big(f(x;\theta_q)\big)\big] \big)^2 \big] = 0 \\
&\Leftrightarrow \forall x \sim \mathcal{P}(\text{X}), \ \mathbb{E}_{y\sim\mathcal{P}(\text{Y}|x)} \big[ 2 \times \big( z(x;\theta_z) - y \cdot \kappa\big(f(x;\theta_q)\big) \big) \big] = 0 \\
&\Leftrightarrow \forall x \sim \mathcal{P}(\text{X}), \ z(x;\theta_z) = \mathbb{E}_{y\sim\mathcal{P}(\text{Y}|x)} \Big[ y \cdot \kappa\big(f(x;\theta_q)\big) \Big] = \mathbb{E}_{y\sim\mathcal{P}(\text{Y}|x)}[y] \cdot \kappa\big(f(x;\theta_q)\big) \\
&\Leftrightarrow \forall x \sim \mathcal{P}(\text{X}), \ \hat{y}|x = \frac{z(x;\theta_z)}{\kappa\big(f(x;\theta_q)\big)} = \mathbb{E}_{y\sim\mathcal{P}(\text{Y}|x)}[y].
\end{aligned}
\tag{22}
$$

### B.5 Discussion for the theoretical correctness of the unbiasedness metrics (TRE and MRE)

Details are shown in Appendix D.2.2.

### B.6 Utilizing GTS for direct probabilistic modeling

Details are shown in Appendix A.5.

## C  Challenges of applying external bias correction approaches in industrial recommender systems

Applying *external* debiasing paradigms from prior works to real-world recommender systems necessitates the development of an additional online post-processing pipeline after the ranking model's scoring pipeline in the overall online system, which suffers from practical challenges including high development costs, substantial maintenance difficulties, and significant system resource consumption. In addition, since the online calibration pipeline [10] has been commonly utilized in many large-scale recommender systems, an alternative approach is to just let this pipeline itself fit the correction. However, in practice, when the retransformation bias problem becomes severe (*e.g.* TRE is about 0.6 before calibration), this alternative approach is prone to yield unstable online TRE results after the calibration, which are significantly worse than those of our TranSUN method, with the mean being 180% and the standard deviation being 150% of ours. The instability of calibration results might be caused by the excessively large differences between predicted values and true values, which can even affect the robustness of the calibration pipeline. Therefore, as a practical solution, we propose TranSUN and Generalized TranSUN (GTS), an end-to-end in-model correction method to address the challenges of existing approaches and provide robust model performance of unbiasedness.

# D  Details in Experimental Setting

## D.1  Detailed setting on synthetic data

### D.1.1  Data and metric

We adopt 8 distributions $\mathcal{H}(Y)$ from three categories: right-skewed, left-skewed, and symmetric. Specifically, we select some classic exponential family instances (*e.g.* Gamma, Beta, Normal), multimodal distributions (*e.g.* bimodal uniform), and industry-prevalent zero-inflated distributions (*e.g.* ZILN [40] with positive rate $p_+ = 0.2$). Detailed implementations are shown in Code 2. For each distribution, we randomly sample $N = 10^6$ instances. Fig. 2 illustrates the density and some statistics of the sampled $\mathcal{P}_s(Y|x)$. We adopt SRE (Signed Relative Error) as the metric, *i.e.* $(\hat{y} - y)/y$ where $\hat{y}$ and $y$ are prediction and ground truth, respectively.

### D.1.2  Implementation details

We implement all methods using the Tensorflow toolbox on an M2 Pro CPU with 16 GB. Since the model input $x$ is fixed in $\mathcal{P}_s(Y|x)$, we directly set it to 1.0. We also implement some advanced classification-based models [23, 38, 21, 9] for comparison, where only their main classification losses are retained. For TPM [23] and CREAD [38], bin number is set to 4, which is produced via equal-frequency and EAD binning method [38], respectively. MDME [21] adopts an equal-width binning method to determine 2 sub-distributions, each containing 2 buckets. WLR [9] is only evaluated on RS-ZIG since it requires substantial zero values as negative samples. We also implement ZILN [40] and OptDist [41], which discard classification branches when not evaluated on RS-ZIG. And OptDist is set to adopt 3 SDNs. We set batch size to 1024 and select the optimal learning rate from $\{10^{-i}\}_{1 \le i \le 3} \cup \{5 \times 10^{-i}\}_{1 \le i \le 4}$. All models are trained for 1 epoch, with each training repeated 10 times with different random seeds to obtain statistical significance, where the $p$-value is calculated by the t-test.

## D.2  Detailed setting on real-world data

### D.2.1  Datasets

We adopt two public datasets (**CIKM16**, **DTMart**) and one exclusive industrial dataset (*abbr.* **Indus**). Specifically, CIKM16 aims to predict the dwell time for each session in online searching. We only use the items in the session as input features. The CIKM16 dataset contains 310,302 sessions and 122,991 items. DTMart aims to predict the transaction amount for each order in marketing activities. We use item ID, categories, time taken for procurement, and delivery as input features. The DTMart dataset contains 985,033 orders and 15,546 items. Indus aims to predict the Gross Merchandise Value (GMV) of each order paid in the recommendation scenario of an anonymous e-commerce platform. The Indus dataset contains 132,657,136 orders, 14,538,759 items, and 67,729,869 users. CIKM16 and DTMart datasets are both randomly divided into training and testing samples at a ratio of 4:1, and the Indus dataset is divided at a ratio of 5:1. Interestingly, we found that the Pareto phenomenon occurred in the Indus dataset, where 80% of the GMV was contributed by the top 30% high-priced orders. Consequently, we further conduct evaluation solely on the top high-priced orders in the testing set, *abbr.* **Indus Top** test, to better analyze model performance.

### D.2.2  Metrics for assessing unbiasedness

As mentioned in Sec. 5.1, we propose Total Ratio Error (TRE) and Mean Ratio Error (MRE) to assess unbiasedness. These metrics are formulated as

$$\text{TRE} = \left| \frac{\sum_{i=1}^{M} \hat{y}_i - y_i}{\sum_{i=1}^{M} \hat{y}_i} \right|, \tag{23}$$

$$\text{MRE} = \left| \frac{1}{M} \sum_{i=1}^{M} \frac{\hat{y}_i - y_i}{\hat{y}_i} \right|, \tag{24}$$

where $\hat{y}_i$ is prediction, $y_i$ is ground truth, and $M$ is sample number.

Firstly, we theoretically prove that achieving either the optimal TRE or MRE metric is a *necessary condition* for the model to be unbiased. Specifically, through the definition of the model's unbiasedness, we can derive that

$$\forall x \sim \mathcal{P}(X), \ \hat{y}(x) = \mathbb{E}_{y \sim \mathcal{P}(Y|x)}[y] \ \Leftrightarrow \ \forall x \sim \mathcal{P}(X), \ \mathbb{E}_{y \sim \mathcal{P}(Y|x)}\big[\hat{y}(x) - y\big] = 0, \quad (25)$$

where $\hat{y}(x) = \hat{y}|x$. For the TRE metric, optimizing it means that

$$
\begin{aligned}
&\min \text{TRE} \\
\Leftrightarrow \ & \left| \frac{\mathbb{E}_{(x,y) \sim \mathcal{D}}\big[\hat{y}(x) - y\big]}{\mathbb{E}_{(x,y) \sim \mathcal{D}}[\hat{y}(x)]} \right| = 0 \\
\Leftrightarrow \ & \mathbb{E}_{(x,y) \sim \mathcal{D}}\big[\hat{y}(x) - y\big] = 0 \\
\Leftrightarrow \ & \mathbb{E}_{x \sim \mathcal{P}(X)}\Big[\mathbb{E}_{y \sim \mathcal{P}(Y|x)}\big[\hat{y}(x) - y\big]\Big] = 0 \\
\Leftarrow \ & \forall x \sim \mathcal{P}(X), \ \mathbb{E}_{y \sim \mathcal{P}(Y|x)}\big[\hat{y}(x) - y\big] = 0,
\end{aligned}
\quad (26)
$$

where the second row in Eq. (26) is derived from the definition of TRE using expectation formulation, and the last row comes from Eq. (25). Regarding the MRE metric, optimizing it implies that

$$
\begin{aligned}
&\min \text{MRE} \\
\Leftrightarrow \ & \left| \mathbb{E}_{(x,y) \sim \mathcal{D}}\Big[1 - \frac{y}{\hat{y}(x)}\Big] \right| = 0 \\
\Leftrightarrow \ & \mathbb{E}_{x \sim \mathcal{P}(X)}\Big[\mathbb{E}_{y \sim \mathcal{P}(Y|x)}\Big[\frac{\hat{y}(x) - y}{\hat{y}(x)}\Big]\Big] = 0 \\
\Leftrightarrow \ & \mathbb{E}_{x \sim \mathcal{P}(X)}\Big[\frac{1}{\hat{y}(x)} \cdot \mathbb{E}_{y \sim \mathcal{P}(Y|x)}\big[\hat{y}(x) - y\big]\Big] = 0 \\
\Leftarrow \ & \forall x \sim \mathcal{P}(X), \ \mathbb{E}_{y \sim \mathcal{P}(Y|x)}\big[\hat{y}(x) - y\big] = 0,
\end{aligned}
\quad (27)
$$

where the second row in Eq. (27) is derived from the definition of MRE using expectation formulation, and the last row comes from Eq. (25). Notably, the derivation in Eq. (27) explains the reason that the model prediction $\hat{y}(x)$ is placed in the denominator position of the metric. Specifically, once the MRE is modified to swap the numerator and denominator of each item in it, as

$$\text{MRE}^{(\text{wr})} = \left| \frac{1}{M} \sum_{i=1}^{M} \frac{y_i - \hat{y}_i}{y_i} \right|, \quad (28)$$

then this new metric is no longer suitable for measuring the model unbiasedness, which actually turns to a necessary condition for the estimation of $1/\mathbb{E}_{y \sim \mathcal{P}(Y|x)}\big[\frac{1}{y}\big]$, as

$$
\begin{aligned}
&\min \text{MRE}^{(\text{wr})} \\
\Leftrightarrow \ & \left| \mathbb{E}_{(x,y) \sim \mathcal{D}}\Big[1 - \frac{\hat{y}(x)}{y}\Big] \right| = 0 \\
\Leftrightarrow \ & \mathbb{E}_{x \sim \mathcal{P}(X)}\Big[\mathbb{E}_{y \sim \mathcal{P}(Y|x)}\Big[1 - \frac{\hat{y}(x)}{y}\Big]\Big] = 0 \\
\Leftrightarrow \ & \mathbb{E}_{x \sim \mathcal{P}(X)}\Big[\hat{y}(x) \cdot \mathbb{E}_{y \sim \mathcal{P}(Y|x)}\Big[\frac{1}{\hat{y}(x)} - \frac{1}{y}\Big]\Big] = 0 \\
\Leftarrow \ & \forall x \sim \mathcal{P}(X), \ \mathbb{E}_{y \sim \mathcal{P}(Y|x)}\Big[\frac{1}{\hat{y}(x)} - \frac{1}{y}\Big] = 0 \\
\Leftrightarrow \ & \forall x \sim \mathcal{P}(X), \ \hat{y}(x) = \frac{1}{\mathbb{E}_{y \sim \mathcal{P}(Y|x)}\big[\frac{1}{y}\big]},
\end{aligned}
\quad (29)
$$

where the second row in Eq. (29) is derived from the definition of $\text{MRE}^{(\text{wr})}$ using expectation formulation.

Next, we theoretically prove that achieving the optimal NRMSE metric is a *necessary and sufficient condition* for the model to be unbiased. Specifically, NRMSE is defined as

$$\text{NRMSE} = \frac{\sqrt{\sum_{i=1}^{M}(\hat{y}_i - y_i)^2}}{\sum_{i=1}^{M} y_i}, \quad (30)$$

and thus optimizing it means that

$$
\begin{aligned}
&\min \mathrm{NRMSE} \\
&\Leftrightarrow \min \mathbb{E}_{(x,y)\sim\mathcal{D}}\big[\big(\hat{y}(x)-y\big)^2\big] \\
&\Leftrightarrow \min \mathbb{E}_{x\sim\mathcal{P}(\mathrm{X})}\big[\mathbb{E}_{y\sim\mathcal{P}(\mathrm{Y}|x)}\big[(\hat{y}(x)-y)^2\big]\big] \\
&\Leftrightarrow \forall x\sim\mathcal{P}(\mathrm{X}),\ \min \mathbb{E}_{y\sim\mathcal{P}(\mathrm{Y}|x)}\big[(\hat{y}(x)-y)^2\big] \\
&\Leftrightarrow \forall x\sim\mathcal{P}(\mathrm{X}),\ \frac{\partial}{\partial\hat{y}(x)}\mathbb{E}_{y\sim\mathcal{P}(\mathrm{Y}|x)}\big[(\hat{y}(x)-y)^2\big]=0 \\
&\Leftrightarrow \forall x\sim\mathcal{P}(\mathrm{X}),\ \mathbb{E}_{y\sim\mathcal{P}(\mathrm{Y}|x)}\big[2\times\big(\hat{y}(x)-y\big)\big]=0 \\
&\Leftrightarrow \forall x\sim\mathcal{P}(\mathrm{X}),\ \hat{y}(x)=\mathbb{E}_{y\sim\mathcal{P}(\mathrm{Y}|x)}[y],
\end{aligned}
\tag{31}
$$

where the second row in Eq. (31) is derived from the definition of NRMSE using expectation formulation, and the last row indicates that $\hat{y}(x)$ is an unbiased estimate.

**Discussion on not using NRMSE as the primary metric for observing unbiasedness.** Since NRMSE is a necessary and sufficient condition for a model to be unbiased, intuitively it should be preferred over TRE and MRE as the primary metric for observing unbiasedness. However, the NRMSE metric has a significant issue, that its optimal value (*i.e.* the minimum) is unknown. This problem hinders us from intuitively perceiving the presence and severity of the bias, making it much difficult to verify the capability of different models in model iterations towards eradicating bias. In contrast, although TRE and MRE are both necessary conditions for a model to be unbiased, their optimal values (*i.e.* the minimum) are known, which is $0.0$. Therefore, they can both clearly indicate whether a model is biased. In our experiments, we take the approach for analyzing unbiasedness where TRE and MRE are initially compared to determine if the model is biased, and then NRMSE is subsequently compared to identify the model with the best unbiasedness. The idea of the analytical approach above is also reflected in our result reporting. For example, NRMSE results are not reported in Table 3 and Fig. 3b, 3c, 3d, which is because the gaps among methods in TRE and MRE are both quite pronounced already.

### D.2.3 Implementation details

For public datasets, all methods employ an "Embedding+MLP" architecture, where continuous features are discretized, each feature's embedding dimension is set to 16, and the MLP has 4 layers with hidden dimensions of 123, 64, and 32. For Indus, all methods employ a DIN architecture [49], where continuous features are also discretized, and the MLP has 4 layers with hidden dimensions of 1024, 512, and 256. The bias modeling branch of TranSUN defaults to non-shared parameters on public datasets and shares the embedding layer on Indus. The default setting for $\epsilon$ is 1.0. For other baselines, WLR [9] follows the implementation in [45]. ZILN [40] and OptDist [41] both discard classification branches. The number of bins in TPM [23] and CREAD [38] is set to 16. MDME [21] adopts 4 sub-distributions, each containing 4 buckets, and OptDist adopts 3 SDNs. The equal-frequency binning method is adopted for both TPM and MDME, while the EAD method is adopted for CREAD. On public datasets, the SGD optimizer is utilized with a batch size of 512 and a learning rate of 0.001. On the Indus, the AdagradDecay optimizer is employed with a batch size of 64 and a learning rate of 0.01. All models adopt only one training epoch, and each training is repeated 5 times with different random seeds to obtain statistical significance, where the $p$-value is calculated by the t-test. We implement all methods using the Tensorflow toolbox. The model training on public datasets is performed on an M2 Pro CPU with 16GB, while the training on the Indus dataset consumes resources including 2000 CPUs with 90 GB. In addition, the statistical significance in the results of the online A/B test (Table 9) is calculated by the z-test.

## E More Experimental Results

### E.1 Biasedness comparison of TPM with different bin numbers on synthetic data

As shown in Table 2, TPM [23] underestimates the mean on left-skew and overestimates it on right-skew data, likely because it learns the average of quantiles while the bin number is only 4. The bin number is set so small to amplify the theoretical biasedness problem of TPM, which can be

significantly alleviated by increasing the bin number, as shown in Table 14. This conclusion also applies to CREAD [38].

Table 14: Comparison results of TPM with different bin numbers on synthetic data, where "$X\#$bin" indicates the bin number of TPM is $X$. Bold indicates the **best** results, and gray background indicates that |SRE| <1% .

| Models | RS-G | RS-BU | RS-ZIG | LS-B | LS-BU | SM-U | SM-TN | SM-BU |
|---|---|---|---|---|---|---|---|---|
| TPM (4#bin) [23] | 0.8303 | 0.1670 | 3.4244 | -0.0411 | -0.0303 | 0.0055 | 0.0080 | 0.0060 |
| TPM (32#bin) [23] | 0.1099 | 0.0519 | 0.4874 | -0.0094 | -0.0157 | 0.0021 | 0.0020 | 0.0033 |
| TPM (256#bin) [23] | **0.0281** | **0.0124** | **0.0561** | **-0.0022** | **-0.0054** | **0.0013** | **0.0013** | **0.0021** |

## E.2 Details for online A/B test

In Taobao App (DAU > 300M), our **LogSUN** model is deployed as GMV prediction models in both product and short video recommendation scenarios in *Guess What You Like* domain in the homepage, which serve as pointwise ranking models for GMV target in the ranking stage of the recommender systems to optimize object distribution. Notably, in the short video recommendation scenario on the e-commerce platform, all the short videos are product description videos, each of which links to a certain product, therefore, the GMV of a short video actually refers to the transaction value of the product described in the video after the short video is clicked.

In the A/B test, our LogSUN is compared against the previous online base (LogMAE) and the commonly used LogMSE. Two common metrics, GMV and GMVPI (GMV Per Impression), are adopted to assess online performance. Due to the large variability of these GMV-related metrics, we allocated either substantial traffic or a sufficiently long period for A/B tests to obtain statistically significant results. Specifically, we conduct two A/B tests for one week in the product recommendation scenario, with LogSUN serving 15% of the total traffic, LogMAE 20%, and LogMSE 9%. And we conduct one A/B test for one month in the short video recommendation scenario, with LogSUN serving 6% of the total traffic, and LogMAE 6%.

As shown in Table 9, LogSUN achieves significant gains (*e.g.* **> +1.0%** GMV gains) over the baselines in both recommendation scenarios. We attribute the consistent gains in online metrics to the following two reasons. The first reason is that LogSUN significantly improves the *predictive accuracy* of samples in each target bin by eradicating the retransformation bias, as shown in Figure 3c. The second is that LogSUN empirically improves the *ranking accuracy* of the top high-priced samples (*e.g.* the improvements in AUC on Indus Top test, as shown in Table 8), which contribute 80% of the total GMV though they are small in number. On the other hand, we observed that the GMV gain in the short video recommendation scenario was much larger than that of the product. This is possibly because the retransformation bias problem is much more severe in the short video recommendation scenario (TRE is about 0.6352, nearly 140% of the product's). Now, LogSUN has been deployed in the two core recommendation scenarios to serve the major traffic.

## E.3 Comparison of GTS instance models on synthetic data

Some common $\mathcal{L}_{\mathcal{H}_q}$ and $\kappa$ are selected to devise different instances of GTS, providing reference results for its application. As exhibited in Table 15, in terms of $\mathcal{L}_{\mathcal{H}_q}$, MSE loss is the most versatile loss as it fastly converges to the unbiased solution in all distributions, whose $\mathbb{Q}$ corresponds to the arithmetic mean point. Among other losses, MAE presents better convergence on right-skewed data, while MSPE and MAPE both converge more effectively on left-skewed and symmetric data. Regarding function $\kappa$, all candidates exhibit unbiasedness consistent with TranSUN. The performance of $\kappa$ on real-world data is shown in Table 10.

## E.4 Architectural sensitivity on real-world data

Since our method introduces an extra bias learning branch to transformed MSE, we explore the impact of different parameter sharing schemes between the added branch and the original model, which is conducted on the LogSUN model. In Table 16, sharing parameters from bottom layers (*e.g.* embedding layers) has minimal impact on performance, whereas further sharing more parameters

Table 15: Comparison results of different GTS instances ($\mathrm{T}(y) = \ln(y+1)$) on synthetic data . Red boldface indicates that **|SRE| >1%**.

| $\mathcal{L}_{\mathcal{H}_q}$ | $\kappa(\cdot)$ | RS-G | RS-BU | RS-ZIG | LS-B | LS-BU | SM-U | SM-TN | SM-BU |
|---|---|---|---|---|---|---|---|---|---|
| MSE | $\kappa(u) = 1/(|\mathrm{T}^{-1}(u)| + \epsilon)$ | 0.0014 | 0.0013 | -0.0010 | 0.0014 | 0.0033 | -0.0038 | -0.0017 | -0.0058 |
| MAE | $\kappa(u) = 1/(|\mathrm{T}^{-1}(u)| + \epsilon)$ | 0.0042 | 0.0010 | -0.0088 | -0.0012 | **0.0242** | **0.0177** | 0.0083 | 0.0010 |
| MSPE | $\kappa(u) = 1/(|\mathrm{T}^{-1}(u)| + \epsilon)$ | 0.0083 | **0.0340** | -0.0048 | -0.0064 | -0.0044 | -0.0074 | 0.0059 | 0.0023 |
| MAPE | $\kappa(u) = 1/(|\mathrm{T}^{-1}(u)| + \epsilon)$ | **0.0103** | **0.0191** | -0.0018 | -0.0087 | -0.0016 | -0.0084 | 0.0047 | -0.0061 |
| MSE | $\kappa(u) = 1/(|u| + \epsilon)$ | 0.0055 | 0.0044 | -0.0011 | -0.0020 | -0.0032 | -0.0034 | 0.0036 | 0.0041 |
| MSE | $\kappa(u) = |u|$ | 0.0018 | 0.0024 | -0.0057 | -0.0014 | -0.0018 | -0.0028 | -0.0012 | -0.0044 |

within MLP results in a noticeable performance degradation, which might be inferred that the added task primarily affects the downstream MLP-modeled fitting function rather than the distribution of latent embedding.

Table 16: Comparison results of the LogSUN model under different parameter sharing architectures on CIKM16 test. "$X/Y$" denotes sharing the first $X$-th layers of the network with totally $Y$ layers. The **best** results are in boldface and the second best are underlined.

| Parameter sharing schemes | TRE $\downarrow$ | MRE $\downarrow$ | NRMSE $\downarrow$ | NMAE $\downarrow$ | XAUC $\uparrow$ |
|---|---|---|---|---|---|
| 0/1 Embedding + 0/3 MLP | **0.0123** | 0.0439 | 0.5625 | 0.4333 | 0.6740 |
| 1/1 Embedding + 0/3 MLP | 0.0147 | **0.0353** | **0.5624** | **0.4305** | **0.6743** |
| 1/1 Embedding + 1/3 MLP | 0.0316 | 0.0413 | 0.5672 | 0.4356 | 0.6707 |
| 1/1 Embedding + 2/3 MLP | 0.0490 | 0.0686 | 0.5684 | 0.4455 | 0.6702 |
| 1/1 Embedding + 3/3 MLP | 0.1063 | 0.0955 | 0.5951 | 0.4775 | 0.6658 |

### E.5 Comparison of GTS instances with different $\kappa$ functions on real-world data

As shown in Table 10, compared to the one used in TranSUN, other $\kappa$ candidates all bring about poorer loss convergence, which is probably because the variances of their $\kappa \cdot y$ are all too large (Table 17), leading to an excessively steep optimization landscape for $\theta_z$.

Table 17: Comparison of the variances of $\kappa \cdot y$ in GTS instances with different $\kappa$ functions ($\mathrm{T}(y) = \ln(y+1)$ and $\mathcal{L}_{\mathcal{H}_q}$ is set to MSE loss) on CIKM16 train.

| $\kappa(\cdot)$ | $\mathrm{Var}(\kappa \cdot y)$ | $\mathrm{Var}(\kappa \cdot y)/(\mathbb{E}[\kappa \cdot y])^2$ |
|---|---|---|
| $\kappa(u) = 1/(|\mathrm{T}^{-1}(u)| + \epsilon)$ | **0.6329** | **0.3610** |
| $\kappa(u) = 1/(|u| + \epsilon)$ | 417.67 | 0.3829 |
| $\kappa(u) = |u|$ | 461790.3 | 0.4447 |

## F Broader Impacts

We think that there is no societal impact of the work performed. Though our method has been applied in two core recommendation scenarios on a leading e-commerce platform (DAU > 300M) to serve the major online traffic, there are no obvious paths that lead to potential harm, since our work just presents an unbiased regression methodology and is not tied to particular applications.

**Code 1:** TensorFlow-style implementation for our TranSUN and GTS.

```python
import numpy as np
import tensorflow.compat.v1 as tf

tf.disable_v2_behavior()
from tensorflow.keras.layers import Dropout

def transun_model(x, y_label, mlp_units, trans,
                  dropout, feat_num, feat_dim, rmask, imask, reuse
                    , is_training,
                  **kwargs):
    with tf.variable_scope('transun', reuse=reuse):
        # embed & pool
        embedding_param = tf.get_variable(name="embtable", shape=[
            feat_num, feat_dim])
        feas = tf.nn.embedding_lookup(ids=x, params=
            embedding_param)  # (bsz,20,feat_dim)
        requests = tf.reduce_sum(feas[:, :10, :] * tf.expand_dims(
            rmask, -1), 1) / tf.reduce_sum(rmask, 1, keepdims=True)
              # avg pooling
        items = tf.reduce_sum(feas[:, 10:, :] * tf.expand_dims(
            imask, -1), 1) / tf.reduce_sum(imask, 1, keepdims=True)
        feas = tf.concat([requests, items], 1)  # (bsz,2*feat_dim)
        # mlp, main branch
        drpt = Dropout(rate=dropout)
        fc = feas
        for idx, num_unit in enumerate(mlp_units):
            fc = linear(fc, num_unit, f"dense{idx}", reuse)
            fc = tf.nn.relu(fc)
            fc = drpt(fc, training=is_training)
        logits = linear(fc, 1, "output", reuse)
        # mlp, sun branch
        fc1 = feas
        for idx, num_unit in enumerate(mlp_units):
            fc1 = linear(fc1, num_unit, f"sun{idx}", reuse)
            fc1 = tf.nn.relu(fc1)
            fc1 = drpt(fc1, training=is_training)
        sun = linear(fc1, 1, "sun_out", reuse)

        bias = 1.0

        if trans == 'log':
            ori_pred = tf.math.exp(tf.math.minimum(10.0,tf.nn.relu
                (logits))) - 1
            final_pred = (ori_pred + bias) * tf.nn.relu(sun)
        elif trans == 'sqrt':
            ori_pred = tf.nn.relu(logits) ** 2
            final_pred = (ori_pred + bias) * tf.nn.relu(sun)
        elif trans == 'atan':
            ori_pred = tf.math.tan(tf.clip_by_value(logits, 0.0,
                np.atan(10000)))
            final_pred = (ori_pred + bias) * tf.nn.relu(sun)
        elif trans == 'tanh':
            ori_pred = tf.math.atanh(tf.clip_by_value(logits, 0.0,
                np.tanh(800)))
            final_pred = (ori_pred + bias) * tf.nn.relu(sun)
        else:
            raise NotImplementedError

        FUSE_THRES = 54.0
        fused_pred = tf.where(ori_pred < FUSE_THRES, ori_pred,
            final_pred)
```

```python
            if is_training:
                if trans == 'log':
                    sun_label=tf.stop_gradient(y_label / (tf.exp(
                        logits) - 1 + bias))
                    mse_loss = tf.reduce_sum(tf.square(tf.log(y_label
                        + 1) - logits))
                    sun_loss = tf.reduce_sum(tf.square(sun - sun_label
                        ))
                elif trans == 'sqrt':
                    sun_label = tf.stop_gradient(y_label / (logits **
                        2 + bias))
                    mse_loss = tf.reduce_sum(tf.square(tf.math.sqrt(
                        y_label) - logits))
                    sun_loss = tf.reduce_sum(tf.square(sun - sun_label
                        ))
                elif trans == 'atan':
                    sun_label=tf.stop_gradient(
                        y_label / (tf.math.tan(tf.clip_by_value(logits
                            , 0.0, np.atan(10000))) + bias))
                    mse_loss = tf.reduce_sum(tf.square(tf.math.atan(
                        y_label) - logits))
                    sun_loss = tf.reduce_sum(tf.square(sun - sun_label
                        ))
                elif trans == 'tanh':
                    sun_label = tf.stop_gradient(y_label / (tf.math.
                        atanh(tf.clip_by_value(logits, 0.0, np.tanh
                        (800))) + bias))
                    mse_loss = tf.reduce_sum(tf.square(tf.math.tanh(
                        y_label) - logits))
                    sun_loss = tf.reduce_sum(tf.square(
                        sun - sun_label)
                        )
                else:
                    raise NotImplementedError

                rank_loss0 = _rank_loss(sun,sun_label)
                rank_loss1 = _rank_loss(logits, y_label)
                rank_loss = rank_loss0 #+ rank_loss1
                return {'loss': mse_loss
                                + sun_loss
                                + rank_loss,
                        'mse_loss': mse_loss,
                        'sun_loss': sun_loss,
                        # 'rank_loss':rank_loss,
                        'gt_mean': tf.reduce_mean(y_label),
                        'sun_gt_mean': tf.reduce_mean(sun_label),
                        'sun_pred_mean': tf.reduce_mean(sun),
                        'pred_mean': tf.reduce_mean(final_pred),
                        'ori_pred_mean': tf.reduce_mean(ori_pred),

                        'pred': final_pred,
                        'gt': y_label,
                        'ori_pred': ori_pred
                        }
            else:
                return {"pred": final_pred,
                        "fused_pred": fused_pred,
                        "ori_pred": ori_pred}

def gts_model(x, y_label, mlp_units,
                dropout, feat_num, feat_dim, rmask, imask, reuse,
                is_training, location='mse', k_func=0, trans='
                log', **kwargs):
    with tf.variable_scope('gts', reuse=reuse):
```

```
05          embedding_param = tf.get_variable(name="embtable", shape=[
                feat_num, feat_dim])
06          feas = tf.nn.embedding_lookup(ids=x, params=
                embedding_param)  # (bsz,20,feat_dim)
07          requests = tf.reduce_sum(feas[:, :10, :] * tf.expand_dims(
                rmask, -1), 1) / tf.reduce_sum(rmask, 1, keepdims=True)
                # avg pooling
08          items = tf.reduce_sum(feas[:, 10:, :] * tf.expand_dims(
                imask, -1), 1) / tf.reduce_sum(imask, 1, keepdims=True)
09          feas = tf.concat([requests, items], 1)  # (bsz,2*feat_dim)
10          # mlp, main branch
11          drpt = Dropout(rate=dropout)
12          fc = feas
13          for idx, num_unit in enumerate(mlp_units):
14              fc = linear(fc, num_unit, f"dense{idx}", reuse)
15              fc = tf.nn.relu(fc)
16              fc = drpt(fc, training=is_training)
17          logits = linear(fc, 1, "output", reuse)
18          fc = feas
19          for idx, num_unit in enumerate(mlp_units):
20              fc = linear(fc, num_unit, f"dense_sun{idx}", reuse)
21              fc = tf.nn.relu(fc)
22              fc = drpt(fc, training=is_training)
23          sun = linear(fc, 1, "sun", reuse)
24
25          assert location in ['mse', 'mae', 'mspe', 'mape']
26          assert k_func in [0, 1, 2]
27          assert trans in ['log']
28
29          if trans == 'log':
30              label = tf.log(y_label + 1)
31              ori_pred = tf.exp(logits) - 1
32
33          if location == 'mse':
34              location_loss = tf.reduce_mean(tf.square(logits -
                    label))
35          elif location == 'mae':
36              location_loss = tf.reduce_mean(tf.abs(logits - label))
37          elif location == 'mspe':
38              location_loss = tf.reduce_mean(tf.square((logits -
                    label) / (label + 1)))
39          elif location == 'mape':
40              location_loss = tf.reduce_mean(tf.abs((logits - label)
                    / (label + 1)))
41
42          if k_func == 0:
43              sun_label = tf.stop_gradient(y_label * tf.exp(-logits)
                    )
44              y_pred = tf.exp(tf.math.minimum(10.0,logits)) * tf.nn.
                    relu(sun)
45          elif k_func == 1:
46              sun_label = tf.stop_gradient(y_label / (1 + tf.abs(
                    logits)))
47              y_pred = (1 + tf.abs(logits)) * tf.nn.relu(sun)
48          elif k_func == 2:
49              sun_label = tf.stop_gradient(y_label * tf.abs(logits))
50              y_pred = tf.nn.relu(sun) / (1e-12+tf.abs(logits))
51          sun_loss = tf.reduce_mean(tf.square(sun - sun_label))
52
53          loss = location_loss + sun_loss
54          if is_training:
55              return {'loss': loss,
56                      # +rank_loss,
57                      'location_loss': location_loss,
58                      'sun_loss': sun_loss,
```

```python
59                    # 'rank_loss':rank_loss,
60                    'gt_mean': tf.reduce_mean(y_label),
61                    'sun_gt_mean': tf.reduce_mean(sun_label),
62                    'sun_pred_mean': tf.reduce_mean(sun),
63                    'pred_mean': tf.reduce_mean(y_pred),
64                    'ori_pred_mean': tf.reduce_mean(ori_pred),
65                    # 'grad_main_emb': tf.gradients([sun_loss],
                          embedding_param),
66                    # 'grad_sun_emb': tf.gradients([sun_loss],
                          embedding_param),
67                    # 'grad_main_logit': tf.gradients([sun_loss],
                          logits),
68
69                    'pred': y_pred,
70                    'gt': y_label,
71                    'ori_pred': ori_pred
72                    }
73        else:
74            return {"pred": y_pred, "ori_pred": ori_pred}
75

76
77  def linear(inputs, units, name, reuse=None):
78      with tf.variable_scope(name, reuse=reuse):
79          input_dim = inputs.get_shape()[-1].value
80          weights = tf.get_variable(
81              'weights', shape=[input_dim, units], initializer=
                  he_initializer)
82          biases = tf.get_variable(
83              'biases', shape=[units], initializer=tf.
                  zeros_initializer())
84          outputs = tf.matmul(inputs, weights) + biases
85          return outputs
86

87
88  def _rank_loss(logits, labels):
89      """
90      @params:
91          - logits: (bsz,1)
92          - labels: (bsz,1)
93      """
94      pairwise_logits = logits - tf.transpose(logits)
95      pairwise_mask = tf.greater(labels - tf.transpose(labels), 0)
96      pairwise_logits = tf.boolean_mask(pairwise_logits,
              pairwise_mask)
97      pairwise_psudo_labels = tf.ones_like(pairwise_logits)
98      rank_loss = tf.reduce_mean(tf.nn.
              sigmoid_cross_entropy_with_logits(
99          logits=pairwise_logits,
200          labels=pairwise_psudo_labels
201      ))
202      # set rank loss to zero if a batch has no positive sample.
203      rank_loss = tf.where(tf.is_nan(rank_loss), tf.zeros_like(
              rank_loss), rank_loss)
204      return rank_loss
```

**Code 2:** Python codes for generating synthetic data.

```python
import numpy as np
from scipy.stats import beta, gamma, uniform, skew, truncnorm

n_samples = 1000000
zero_inflate_ratio = 0.8

def generate_distributions():
    distributions = [] # [(title, samples, mean, skewn, var,
        caption)]

    # (a) RS-G
    d = gamma(a=2, scale=1)
    samples = d.rvs(n_samples)
    true_mean = d.mean()
    skewn = d.stats(moments='s')
    var = d.var()
    distributions.append(("Right-Skewed:_Gamma", samples,
        true_mean, skewn, var, '(a)_RS-G'))

    # (b) RS-BU
    low_samples = uniform(loc=1, scale=10).rvs(int(n_samples *
        0.9))
    high_samples = uniform(loc=90, scale=10).rvs(int(n_samples *
        0.1))
    samples = np.concatenate([low_samples, high_samples])
    true_mean = 0.9 * 6 + 0.1 * 95
    skewn = float(skew(samples))
    var = np.var(samples)
    distributions.append(("Right-Skewed:_Bi-Uniform", samples,
        true_mean, skewn, var,'(b)_RS-BU'))

    # (c) RS-ZIG
    base_samples = gamma(a=2, scale=1).rvs(n_samples)
    zero_mask = np.random.rand(n_samples) < zero_inflate_ratio
    samples = base_samples * (1 - zero_mask)
    true_mean = gamma(a=2, scale=1).mean() * (1 -
        zero_inflate_ratio)
    skewn = skew(samples)
    var = np.var(samples)
    distributions.append(("Right-Skewed:_Zero-Inflated_Gamma",
        samples, true_mean, skewn, var,'(c)_RS-ZIG'))

    # (d) LS-B
    d = beta(a=3, b=1.5)
    samples = d.rvs(n_samples)
    true_mean = d.mean()
    skewn = d.stats(moments='s')
    var = d.var()
    distributions.append(("Left-Skewed:_Beta", samples, true_mean,
        skewn, var,'(d)_LS-B'))

    # (e) LS-BU
    low_samples = uniform(loc=1, scale=10).rvs(int(n_samples *
        0.1))
    high_samples = uniform(loc=90, scale=10).rvs(int(n_samples *
        0.9))
    samples = np.concatenate([low_samples, high_samples])
    true_mean = 0.1 * 6 + 0.9 * 95
    skewn = float(skew(samples))
    var = np.var(samples)
    distributions.append(("Left-Skewed:_Bi-Uniform", samples,
        true_mean, skewn, var,'(e)_LS-BU'))
```

```python
        # (f) SM-U
        d = uniform(loc=0, scale=100)
        samples = d.rvs(n_samples)
        true_mean = d.mean()
        skewn = d.stats(moments='s')
        var = d.var()
        distributions.append(("Symmetric:␣Uniform", samples, true_mean
            , skewn, var,'(f)␣SM-U'))

        # (g) SM-TN
        a, b = (0 - 50) / 10, (100 - 50) / 10
        samples = truncnorm(a, b, loc=50, scale=10).rvs(n_samples)
        true_mean = 50
        skewn = skew(samples)
        var = d.var()
        distributions.append(("Symmetric:␣Trunc-Normal", samples,
            true_mean, skewn, var,'(g)␣SM-TN'))

        # (h) SM-BU
        low_samples = uniform(loc=1, scale=10).rvs(int(n_samples *
            0.5))
        high_samples = uniform(loc=90, scale=10).rvs(int(n_samples *
            0.5))
        samples = np.concatenate([low_samples, high_samples])
        true_mean = 0.5 * 6 + 0.5 * 95
        skewn = float(skew(samples))
        var = np.var(samples)
        distributions.append(("Symmetric:␣Bi-Uniform", samples,
            true_mean, skewn, var,'(h)␣SM-BU'))

        return distributions

dist = generate_distributions()
```

