# OpenReview forum: "TranSUN: A Preemptive Paradigm to Eradicate Retransformation Bias Intrinsically from Regression Models in Recommender Systems"
_NeurIPS.cc/2025/Conference — NeurIPS 2025 poster_

### Official Review · Reviewer_xLUq · 2025-06-30

**Clarity:** 3
**Significance:** 3
**Originality:** 3
**Rating:** 5
**Confidence:** 5

**Summary:**

This paper proposes a novel preemptive paradigm called TranSUN to address the long-neglected retransformation bias problem in regression models for recommender systems. TranSUN introduces a joint bias learning approach, ensuring theoretically guaranteed unbiasedness and superior empirical convergence. The paper also generalizes TranSUN into a generic regression model family, Generalized TranSUN (GTS), which offers more theoretical insights and serves as a flexible framework for developing various bias-free models. Extensive experiments demonstrate the superiority and broad applicability of the proposed methods.

**Questions:**

- How proposed method performs w.r.t other ranking metrics (such as NDCG .etc)?
- How to theoretically explain the improvement on pairwise/listwise ranking metrics?
- How to resolve  the aforementioned paradox between the high-skewed distribution and retransformation bias?

**Ethical Concerns:**

["NO or VERY MINOR ethics concerns only"]

**Final Justification:**

The authors' responses have clarified all my concerns. Therefore, I am now inclined to raise the evaluation score for this paper.

**Quality:**

3

**Strengths And Weaknesses:**

Strengths:
- This paper is clearly written and well organized.
- The retransformation bias is a crucial problem but neglected by most recommendation methods, so the motivation of this paper is interesting.
- The experimental results are promising.

Concerns:
- In recommendation tasks, we often use pairwise or listwise ranking metrics (e.g. XAUC adopted by authors) for evaluation.  However, the function T can only affect the labels y in a pointwise manner (Eq.(1) in this paper), but not change their relative ranking orders( E[sign(y_1>y_2)] <=> E[sign(T(y_1)>T(y_2)] ). In this way, I wonder how the retransformation bias affect the model performance w.r.t the ranking metrics. I hope the authors could provide some theoretical justification on this point. Otherwise, we can reasonably speculate that the improvement on AUC does not stem from the proposed strategy, and thus its application scope might be limited.
- By introducing T(y), the model can adapt the high-skewed distribution of y, but this operation leads to a Jensen gap (Eq.(3)). To fill this gap, the author proposes Eq.(9). However, the optimization of L_{sun} in Eq.(9) might also be affected by high-skewedness of y in turn. This seems to be a paradox.

-----!!!

In the rebuttal phase, the response from the authors have resolved all my concerns above. Therefore, I intend to improve the rating of this paper.

---

> ### Author Rebuttal · Authors · 2025-07-30
>
> We thank all the reviewers for their time and insightful feedback to help improve our work. We appreciate that the reviewers recognized our research problem as practically valuable and our method as simple, effective, theoretically grounded, and "of broad practical potential", supported by exhaustive validation. We have carefully considered all the comments and addressed them in this rebuttal, which will be incorporated into the revised version. We would be grateful if you could reconsider our work in light of these explanations and the provided evidence. If any further clarification is required, please do not hesitate to contact us. Below we address your questions (Q) and weaknesses (W).
> > ### W1&Q2: Since retransformation bias (RB) does not change label rank order in the listwise manner, how does it affect XAUC?
>
> Since our model is trained in a pointwise manner, retransformation bias (RB) can affect the rank order of model predictions, which in turn affects the rank metrics. That is, $\mathbb{E}[\mathrm{Y}|x_1]<\mathbb{E}[\mathrm{Y}|x2] \not \Leftrightarrow \mathrm{T}^{-1}(\mathbb{E}[\mathrm{T}(\mathrm{Y})|x1])<\mathrm{T}^{-1}(\mathbb{E}[\mathrm{T}(\mathrm{Y})|x2])$. For clarity, let’s start with a case study, then we clarify the reasons for the improvement/drop of XAUC, and finally introduce how we address the relationship between XAUC and online metrics in real-world industrial recommendation systems.
>
> 1. **Case study**: We first sample two inputs {x1, x2} ~ P(X), and then sample 3 groups of target y from P(Y|x1) and P(Y|x2) respectively, and let the samples be {3,3,3} ~ P(Y|x1) and {8,1,1} ~ P(Y|x2), and let T(x)=log(x). Then:
> 	- The LogMSE prediction is $f(x_1)=\mathrm{T}^{-1}(\mathbb{E}[\mathrm{T}(\mathrm{Y})|x_1])=\exp((\log(3)+\log(3)+\log(3))/3)=3$, $f(x_2)=\mathrm{T}^{-1}(\mathbb{E}[\mathrm{T}(\mathrm{Y})|x_2])=\exp((\log(8)+\log(1)+\log(1))/3)=2$, where $f(x_1)>f(x_2)$ and the bias occurs on the sample X=x2
> 	- The TranSUN/GTS prediction is $f(x_1)=\mathbb{E}[\mathrm{Y}|x_1]=(3+3+3)/3=3, f(x_2)=\mathbb{E}[\mathrm{Y}|x_2]=(1+1+8)/3=10/3$, where $f(x_1)<f(x_2)$
> 	- The descending ranking result of LogMSE is {3, 3, 3, 8, 1, 1}, with **XAUC = 0.333**, NRMSE = 0.324, and TRE = 4/15, while the result of TranSUN/GTS is {8, 1, 1, 3, 3, 3}, with **XAUC = 0.233**, NRMSE = 0.301, and TRE = 0
> 	- It is obviously observed that XAUC metric is affected by the retransformation bias problem
>  2. **Reason for the improvement of high-value XAUC**: TranSUN/GTS compensates underestimation brought by retransformation bias, resulting in better point and rank accuracy of high-value samples  (e.g., for sample $(x_2,8)$, relative error improves from 0.75 to 0.58, and its rank order moves from 4th to 1st, consistent with results on Indus Top(Tab8)), which significantly improves the overall point accuracy (e.g. improvements in NRMSE & TRE in the case) since these metrics are **value-aware** (i.e. metric calculation aware of absolute value of sample label)
>  3. **Reason for the drop of overall XAUC**: Correcting retransformation bias also leads to some hard low-value samples being overestimated (e.g., predictions for the two samples $(x_2,1)$ increase from 2 to 10/3, consistent with Fig3c), which dominates the overall XAUC drop since there are far more hard low-value samples than high-value ones in highly right-skewed data and XAUC metric is **value-agnostic** (i.e. metric calculation only aware of the relative rank order of sample label but not aware to its absolute value)
> 3. **Overall XAUC is inconsistent with online metrics, thus we use high-value XAUC and NDCG**:
> 	- The core goal of a recommender system is to improve online business metrics, while XAUC is merely an intermediate metric. For business target modeled by regression (GMV, watch time, etc.), the event of ranking high-valued samples in the correct position is far more important for online metrics than that of low-valued samples. However, these two events are equally weighted (value-agnostic) in the calculation of XAUC, thereby being dominated by abundant low-valued samples when data are highly right-skewed, which probably contribute only a small portion of online metrics (e.g. 70% of the top low-valued samples contribute only 20% of GMV in Indus). Therefore, overall XAUC is no longer consistent with online metrics
> 	- To address this, we supplemented the offline assessment with a more consistent metric, high-value XAUC (L565-568, Tab8), based on which we achieved improved online results (Tab6). Additionally, NDCG also addresses XAUC's limitation, so we also provide their results (shown in the reply to Q1)
>
> > ### W2&Q3: The optimization of $L_{sun}^{\mathrm{T}}$ in Eq(9) might also be affected by high-skewedness of y, which seems to be a paradox
>
> We beg to differ. **The convergence smoothness of $L_{sun}^{\mathrm{T}}$ is more superior than that of the vanilla MSE (using target y).** This is because the variance and the outlier proportion of the their targets are quite different, resulting in optimization landscapes with different degrees of smoothness. The table below shows the target variance (Var), the variance divided by squared expectation (V/E2), and the target outlier proportion (Outlier%) outside of 3 times the standard deviation of MSE, T-MSE, and $L_{sun}^{\mathrm{T}}$, where T(x)=log(x). As shown, the target of $L_{sun}^{\mathrm{T}}$ has a smaller variance and a better outlier distribution than the vanilla MSE, thus $L_{sun}^{\mathrm{T}}$ converges more smoothly (Fig1). Taking a step back, even if the target formulation of $L_{sun}^{\mathrm{T}}$ encounters convergence difficulties, it can be addressed by optimizing the $\kappa$ function of GTS (L165-168). Therefore, TranSUN/GTS essentially decomposes a hard task into two easier and smoother tasks, rather than the paradox problem you mentioned.
>
> | Loss (CIKM16) | Var | V/E2 | Outlier% |
> |---|:---:|:---:|:---:|
> | MSE | 16440 | 0.41 | 1.03% |
> | $L_{sun}^{\mathrm{T}}$ | 0.63 | 0.36 | 0.59% |
> | LogMSE | 0.62 | 0.02 | 0.29% |
>
> | Loss (Indus) | Var | V/E2 | Outlier% |
> |---|---|---|---|
> | MSE | 97494.16 | 12.10 | 0.78% |
> | $L_{sun}^{\mathrm{T}}$ | 2.327 | 0.93 | 0.32% |
> | LogMSE | 1.87 | 0.15 | 0.28% |
>
> > ### Q1: Performance on NDCG
>
> We provide the NDCG metric results of our TranSUN and other baselines on the CIKM16 test and DTMart test in the table below. NDCG is computed within all samples (NDCG@all) and the top 10% high-value ones (NDCG@10%) respectively. Key observations include:
>  1. TranSUN performs better in NDCG@all than in XAUC, probably because that NDCG is more robust to the low-value samples than XAUC
>  2. TranSUN shows superior performance in NDCG@10%, consistent with the 2nd reason stated in the reply to W1&Q2
>
> | Model | NDCG@all (CIKM16) | NDCG@10%(CIMK16) | NDCG@all (DTMart) | NDCG@10% (DTMart) |
> |---|:---:|:---:|:---:|:---:|
> | MSE | 0.9180 | 0.5201 | 0.9770 | 0.9692 |
> | MAE | 0.9497 | 0.5654 | 0.9867 | 0.9665 |
> | LogMSE | 0.9498 | 0.5517 | 0.9960 | 0.9736 |
> | WLR | 0.9475 | 0.5433 | 0.9693 | 0.9609 |
> | ZILN | 0.9485 | 0.5781 | 0.9963 | 0.9756 |
> | MDME | 0.9391 | 0.5671 | 0.9763 | 0.9639 |
> | TPM | 0.9502 | 0.5888 | 0.9960 | 0.9772 |
> | CREAD | **0.9504** | 0.5881 | 0.9962 | 0.9774 |
> | OptDist | 0.9500 | 0.5852 | 0.9965 | 0.9770 |
> | **LogSUN** | 0.9499 | **0.5904** | **0.9967** | **0.9779** |

---

> > ### Comment · Reviewer_xLUq · 2025-08-06
> > **Discussion**
> >
> > Thanks for the response from the authors. I think all my concerns have been well resolved, and I suggest the authors adding these contents into the original manuscript, clarifying the crucial insights behind the proposed method.

---

> ### Author Response · Authors · 2025-08-05
> **Follow-up Regarding Rebuttal**
>
> Dear Reviewer,
>
> We hope this message finds you well.
>
> Thank you very much for your thoughtful and thorough review of our submission. We truly appreciate the time and effort you have dedicated to providing such constructive feedback.
>
> During the rebuttal period, **we've made every effort to address all of your concerns and questions in detail**. We also incorporated your suggestions, which we believe have helped to further improve the quality and clarity of our paper. If there is **anything else that requires clarification**, we would be more than happy to provide further details.
>
> We completely understand that your schedule is very busy, and we are grateful for the important service you provide to the community. If you have an opportunity to take another look at our responses, we would greatly appreciate any additional feedback or thoughts you might have.
>
> Thank you again for your time and consideration. We look forward to any further comments you may have.
>
> Best regards,
>
> All authors of paper 5113

---

> ### Author Response · Authors · 2025-08-06
> **Reply to Reviewer xLUq**
>
> Thank you sincerely for your positive and encouraging feedback! We are very pleased to know that our responses have addressed your concerns well. As you suggested, **we promise to carefully incorporate these clarifications and highlight the crucial insights behind the proposed method in the revised version of our manuscript**.
>
> We greatly appreciate your thoughtful comments and support throughout the review process, which have been invaluable in helping us improve the quality of our work. *We hope that the resolved concerns and the strengthened manuscript will be reflected favorably in the final evaluation*.
>
> Thank you again for your time, constructive feedback, and encouragement!

---

### Official Review · Reviewer_1oaS · 2025-07-04

**Clarity:** 4
**Significance:** 3
**Originality:** 2
**Rating:** 4
**Confidence:** 4

**Summary:**

This paper focuses on addressing the **re-transformation bias** issue in regression models for recommendation systems. This problem arises when functions like the logarithm are used to transform target values with skewed distributions (e.g., GMV) to stabilize training. When the model's predictions are transformed back to the original scale, a systematic bias is introduced. The authors propose a "proactive" method called **TranSUN**, which introduces a jointly trained auxiliary model branch to learn and correct this bias from the ground up, thereby theoretically guaranteeing an unbiased prediction. This method is further generalized into the **GTS (Generalized TranSUN)** framework, providing a universal approach for designing unbiased regression models. The paper validates the effectiveness of its method through synthetic data, public datasets, and a large-scale online A/B test on a leading e-commerce platform.

**Questions:**

1. **On the Root Cause of Ranking Performance Degradation:** You mentioned that the decrease in XAUC is an acceptable trade-off. Could you please elaborate on why a method designed for more "accurate" predictions leads to a decline in ranking ability? Does this imply that the model, in correcting for large biases on a few high-value samples, sacrifices its ranking discrimination for the majority of samples? Would this limit its general applicability? Additionally, you mentioned that a part of it increased; could you explain what happened there? You state the drop in XAUC is "acceptable given the improvement on TRE" (§5.2). Does this imply unbiasedness hurts ranking? Please analyze the correlation between prediction error reduction (TRE) and ranking metrics (XAUC) to clarify this trade-off.
2. **Comparison with Direct Probabilistic Modeling Methods:** Your method can be viewed as a sophisticated correction to a Gaussian distribution assumption. How do you see it comparing to methods that directly model p(y|x) using more powerful distributional assumptions (e.g., via MDNs or Normalizing Flows)? The latter can obtain unbiased expected predictions in one step without introducing extra tricks like `stop_grad` or auxiliary networks. In your view, what advantages does TranSUN offer over these methods (e.g., in terms of training stability, computational overhead, ease of integration)?
3. Can you prove that the linear transformation assumption (Eq. 11) holds under non-Gaussian noise or multi-modal target distributions (e.g., RS-BU in Fig. 2)? If not, what are the theoretical guarantees when this assumption is violated?
4. **Necessity of the GTS Framework:** You generalized TranSUN to the GTS framework. However, the experimental section mainly validates TranSUN and its variants. Beyond theoretical elegance, what unique and validated practical value does the GTS framework offer beyond TranSUN? Is there a specific scenario where standard TranSUN would fail, and one must leverage the GTS framework to design a specific, non-trivial instance to solve the problem?
5. How much additional training time and computational cost does TranSUN introduce during training? Is this mentioned in the paper? Could you provide an example to give a concrete sense of the scale? Does this impact computational performance and speed in industrial scenarios? The auxiliary branch (Eq. 4) and dynamic slope (Eq. 9) inevitably add computational load. Please quantify: (a) the inference latency (in ms) of TranSUN vs. LogMSE on the Indus dataset, and (b) the extra CPU/GPU resources consumed during online serving at the scale of over 300 million DAU.
6. Are there principled guidelines for determining the optimal $L_v$ (e.g., MAE for right-skewed data) and $\Psi$ functions, or is manual trial-and-error the only way? Does this limit the method's applicability in resource-constrained environments?

**Ethical Concerns:**

["NO or VERY MINOR ethics concerns only"]

**Final Justification:**

The authors have resolved most of my concerns. I decided to keep my score.

**Limitations:**

Yes. The authors discuss limitations in Section 6. However, I believe the discussion of the core limitation—the degradation in ranking performance (XAUC)—is too understated. The authors attribute it to the "no free lunch" theorem but fail to sufficiently explore the severity of this issue. For a paper on recommendation systems, harming the core ranking metric is a very serious problem. This needs to be discussed more explicitly and deeply as a core limitation, with an analysis of how it restricts the method's scope of application.

**Paper Formatting Concerns:**

None.

**Quality:**

3

**Strengths And Weaknesses:**

**Pros:**

1. **Addresses a Practical Problem:** The paper accurately identifies and solves a concrete technical problem prevalent in the industry (especially in e-commerce, video streaming, etc.). Eliminating systematic bias has clear business value for scenarios requiring precise prediction of business metrics (e.g., GMV, user watch time). The paper introduces an auxiliary branch to learn the bias, supervised by a multiplicative bias modeling scheme aimed at capturing the ratio between predicted and true values. This approach tackles the model performance degradation caused by right-skewed data distributions in real-world applications and provides a detailed theoretical foundation.
2. **Exhaustive Experimental Validation:** The experimental work in this paper is very solid, particularly the evaluation on large-scale industrial datasets and the successful deployment in an online A/B test. This directly proves the method's utility and effectiveness in specific business scenarios. This is the most commendable part of the paper. The method is not limited to e-commerce platforms and can be applied to recommendation systems in various other domains, showing its broad practical potential.

**Cons:**

1. **Limited Research Paradigm and Problem Significance:** Although re-transformation bias exists, it is fundamentally a product of the traditional "predict-then-rank" paradigm. Compared to concurrent cutting-edge research in recommendation systems, this paradigm itself is being challenged. Much of the frontier work is shifting towards **end-to-end Learning-to-Rank** frameworks that directly optimize ranking metrics (like NDCG). These frameworks may not rely on precise point estimates of continuous values from the outset, thus naturally circumventing the re-transformation bias problem. From this perspective, the paper's effort to patch a problem within a specific paradigm makes its significance and novelty relatively limited.
2. The study primarily relies on TRE and MRE as core unbiasedness metrics (§5.1), justifying their necessity through the derivations in Appendix C.2.2 (Eq. (21)-(22)). However, these metrics are necessary but not sufficient conditions for unbiasedness: even if they reach their optimal value (zero), it does not guarantee the absence of bias across all data subgroups. For instance, systematic prediction errors might persist in local data distributions undetected by TRE/MRE.
3. **Limited Methodological Innovation:** The core idea of TranSUN—introducing an auxiliary network to learn a correction term—is not uncommon in the broader machine learning field. It can be seen as a direct application of residual fitting, the **boosting idea** from ensemble learning, or an auxiliary task that provides corrections for a primary task in multi-task learning. The authors have skillfully applied it to the specific problem of re-transformation bias, but its methodological originality is relatively modest.
4. **At the Cost of Core Ranking Performance:** This is one of the paper's weaknesses. The paper explicitly states that while the method improves the accuracy of point estimates, it harms the core task of the recommendation system—its ranking ability (as indicated by the drop in the XAUC metric). The authors acknowledge this trade-off but fail to provide an effective solution or an in-depth analysis of its causes, which may make the method unacceptable in the vast majority of recommendation scenarios where ranking performance is the priority.
5. **Gap with Frontier Probabilistic Modeling Methods:** Compared to contemporary research, the paper's reliance on a "transform-correct" method appears somewhat conservative. Currently, using deep probabilistic models like **Mixture Density Networks (MDNs) or Normalizing Flows** to directly model the complex, skewed raw data distribution p(y|x) has become a significant research direction. These methods can directly and unbiasedly compute the conditional expectation E[y|x] and provide richer probabilistic information (such as uncertainty), fundamentally avoiding the re-transformation problem. In contrast, TranSUN/GTS remains a point estimation model, lagging behind these advanced methods in terms of model expressiveness and the richness of its output information.
6. While the paper highlights that TranSUN is deployed on an e-commerce platform with over 300 million daily active users (DAU) (§5.2), it fails to quantify the computational overhead introduced by its key components: the auxiliary branch (Eq. 4) and the dynamic slope (Eq. 9). Crucially, the paper does not provide a comparison of inference latency against baseline methods.

---

> ### Author Rebuttal · Authors · 2025-07-31
>
> We thank all the reviewers for their time and insightful feedback to help improve our work. We appreciate that the reviewers recognized our research problem as practically valuable and our method as simple, effective, theoretically grounded, and "of broad practical potential", supported by exhaustive validation. We have carefully considered all the comments and addressed them in this rebuttal, which will be incorporated into the revised version. If any further clarification is required, please do not hesitate to contact us. Below we address your questions (Q) and weaknesses (W).
>
> > ### W1: Limited research paradigm & problem significance
>
> We beg to differ. ***As currently all large-scale recommender systems adopt "predict-then-rank" paradigm, our research problem on point accuracy has high practical value.*** This is because such platforms serve plentiful business targets (clicks, pays, duration, etc.) & types (products, videos, ads, etc.), which are diverse & frequently updated, making unifying them into an e2e LTR framework impractical. The practical approach is to first model each business type & target separately in ranking stage, then unify them in a upstream re-ranking stage where they compete with each other for traffics. For fair competition, two core demands arise: 1) alignment of values across business types; 2) target score must match real value for physical definition. Both require point accuracy. Under this background, impactful works (e.g. SIR [1] in calibration topic) have emerged in industry.
>
> > ### W2: TRE and MRE are just necessary conditions
>
> This concern is addressed in detail in L588-606. In short, we first examine TRE and MRE to detect model bias, then compare NRMSE (a necessary and sufficient condition) to identify the best unbiased model.
>
> > ### W3: The core idea of TranSUN is residual fitting, whose originality is relatively modest
>
> We beg to differ. Our method's true spirit is **conditional linearity**, i.e. treating the original model $f$ as a conditional slope generator and applying "piecewise" linear transformation on target $y$ in the "residual" branch, which is what we deeply hope readers take away. As discussed in L135-141, bias learning/residual fitting is only a superficial description of TranSUN and does not ensure theoretical unbiasedness. For instance, the $L_{sun}^{\mathrm{T}}$ variant in Eq14 is also residual fitting but is not theoretically unbiased (L502-506, Tab5).
>
> > ### W4&Q1: At the cost of core ranking performance & Root cause of XAUC degradation
>
> First, we clarify that ***the only core goal of recommender system is online business metric***. The observed inconsistency (offline XAUC drops while online metric improves (Tab6)) shows that ***XAUC is no longer suitable for offline evaluation***. Next, we elaborate on the cause of the trade-off between point accuracy and XAUC, and highlight XAUC's limitation. To address this, we adopted two more consistent offline metrics, high-value XAUC (L565-568, Tab 8) and a common rank metric NDCG, on which our model exhibits superior performance. For clarity, we start with a case:
> 1. **Trade-off case**: We first sample inputs {x1, x2} ~ P(X), then sample 3 target y from P(Y|x1) and P(Y|x2) respectively: {3,3,3} ~ P(Y|x1), {8,1,1} ~ P(Y|x2), and let T(x)=log(x). Then:
> 	- LogMSE prediction is $f(x_1)=\mathrm{T}^{-1}(\mathbb{E}[\mathrm{T}(\mathrm{Y})|x_1])=\exp((\log(3)+\log(3)+\log(3))/3)=3$, $f(x_2)=\mathrm{T}^{-1}(\mathbb{E}[\mathrm{T}(\mathrm{Y})|x_2])=\exp((\log(8)+\log(1)+\log(1))/3)=2$; here f(x1)>f(x2), bias on X=x2
> 	- TranSUN/GTS prediction is $f(x_1)=\mathbb{E}[\mathrm{Y}|x_1]=(3+3+3)/3=3, f(x_2)=\mathbb{E}[\mathrm{Y}|x_2]=(1+1+8)/3=10/3$; f(x1)<f(x2)
> 	- LogMSE's descending ranking result is {3, 3, 3, 8, 1, 1}, XAUC = 0.333, NRMSE = 0.324, TRE = 4/15; TranSUN/GTS's is {8, 1, 1, 3, 3, 3}, XAUC = 0.233, NRMSE = 0.301, TRE = 0
> 	- Thus, **a trade-off occurs** : TranSUN/GTS achieves better point accuracy (NRMSE, TRE) while XAUC worsens
>  2. **Root cause**:
> 	 - TranSUN/GTS corrects underestimation from RetransBias, improving point & rank accuracy of high-value samples  (e.g.  (x2,8): relative error 0.75 → 0.58, rank 4th → 1st, as in Tab8 (Indus Top)), which boosts the overall point accuracy (e.g. NRMSE & TRE gains in case) since these metrics are **aware of label's absolute value**
> 	 -  Correction also overestimates some hard low-value samples (e.g. two (x2,1)'s pred rise from 2 to 10/3, as in Fig3c), dominating overall XAUC drop since such samples far outnumber high-value ones in highly right-skewed data and XAUC metric is **only aware of label's relative rank but not of its absolute value**
> 3. **Overall XAUC is inconsistent with online metrics, thus we use high-value XAUC and NDCG**:
> 	- The core goal of recommender system is to improve online metrics, while XAUC is an intermediate metric. For regression targets (GMV, time, etc.), correctly ranking high-value samples matters much more than low-value ones for online metrics. However, XAUC weighs both equally (value-agnostic), so in right-skewed data, low-value samples dominate XAUC but contribute little online (e.g. 70% low-valued samples bring only 20% GMV in Indus). Thus, overall XAUC fails to align with online metrics
> 	- To address this, we supplement with a more consistent metric, high-value XAUC (L565-568, Tab8), based on which we achieved improved online results (Tab6). NDCG also addresses XAUC's limitation, so we further attach its results. In the table below, NDCG is computed for all samples (NDCG@all) and the top 10% high-value ones (NDCG@10%). Key observations: 1) TranSUN performs better in NDCG@all than in XAUC, as NDCG is more robust to low-value samples. 2) TranSUN shows superior performance in NDCG@10%, consistent with the 2nd point
>
> | Model | NDCG@all (CIKM16) | NDCG@10%(CIMK16) | NDCG@all (DTMart) | NDCG@10% (DTMart) |
> |---|:---:|:---:|:---:|:---:|
> | MSE | 0.9180 | 0.5201 | 0.9770 | 0.9692 |
> | MAE | 0.9497 | 0.5654 | 0.9867 | 0.9665 |
> | LogMSE | 0.9498 | 0.5517 | 0.9960 | 0.9736 |
> | WLR | 0.9475 | 0.5433 | 0.9693 | 0.9609 |
> | ZILN | 0.9485 | 0.5781 | 0.9963 | 0.9756 |
> | MDME | 0.9391 | 0.5671 | 0.9763 | 0.9639 |
> | TPM | 0.9502 | 0.5888 | 0.9960 | 0.9772 |
> | CREAD | **0.9504** | 0.5881 | 0.9962 | 0.9774 |
> | OptDist | 0.9500 | 0.5852 | 0.9965 | 0.9770 |
> | **LogSUN** | 0.9499 | **0.5904** | **0.9967** | **0.9779** |
>
> > ### W5&Q2: Gap & Comparison with probabilistic modeling methods (PMM), e.g. MDN, NF
>
> When GTS serves as a point estimator, we acknowledge that PMM offers richer probability information. However:
>
> 1. Inspired by your comment, GTS can, with minor changes (modeling $\sigma$ in Eq11 as a function of $x$), model P(Y|x) in the CTM[2] manner. Specifically, GTS’s assumption becomes:  $-z(x)+Y_x*\kappa(\mathbb{Q})*s(x)\sim N(0,1)$, where $s(x)>0$ is the inverse of the std. GTS can then learn P(Y<v|x) using CTM loss formulas (Eq12 in CLTM[3]). Detailed derivation will be added to the appendix.
>
> 2. As a point estimator, GTS has three advantages over PMM:
>
> 	- **Flexibility**: It can be flexibly plugged into any model with minor revision (L172-176, Tab9). Another advantage is that in industrial practice TranSUN can be fully warmed-up from the online base model, further accelerating TranSUN's convergence
> 	- **Lightweight**: It requires much less resource than PMM. For example, MDN’s overhead grows linearly with mixture components, and NF incurs significant overhead due to Jacobian computations
> 	- **Training stability**: PMM is much harder to converge than GTS given their complex assumptions. For example, on Indus, OptDist[4] (an MDN case) needs 2.13x the iterations TranSUN requires to reach batch-NRMSE=3.0
>
> > ### W6&Q5: Lack of overhead analysis
>
> We provide resource overhead date for the online TranSUN model in the e-commerce platform's recommender system, which achieves a ***high ROI*** (with Tab6):
> 1. **Training**: With 800 workers, LogMSE and TranSUN achieve 89.2 and 86.8 update iterations per second, respectively; thus, TranSUN reduces training speed by only 2.7% and increases training time by 2.78%
> 2. **Inference**: LogMSE has mean/p99 latencies of 27.8ms/49.8ms, while TranSUN’s are 28.1ms/51.6ms
> 3. **Deployment**: On a platform with 300 million DAU, TranSUN required no extra CPU and only 5% more GPU resources
>
> > ### Q3: What are the theoretical guarantees when Eq11 is violated
>
> As noted in L164-165, violating the model assumption does not invalid its intrinsic theoretical unbiasedness. As shown in Tab1, TranSUN and MSE remain theoretically unbiased regardless of the real data distribution (L195-198)
>
> > ### Q4: Necessity of GTS
>
> As a supplement to L169-176, we further discuss GTS's practical value beyond TranSUN.
> 1. TranSUN can only be applied to T-MSE, while GTS to any regression model (Tab9)
> 2. TranSUN may face convergence issues when model assumption is severely violated, though we haven't met this yet. In such cases, one can customize $\mathbb{Q}$ and $\kappa$ to better align its assumption with the real data distribution (L165-168), thereby improving convergence
>
> > ### Q6: Principled guidelines for GTS
>
> The principled guideline is to align the model assumption (Eq11) as closely as possible with the real data, though this requires manual trial-and-error. Our experience is to first fit the data solely using a mainstream regression model as $\mathbb{Q}$, then analyze the skewness and kurtosis of $Y*\kappa(\mathbb{Q})$ to select a suitable $\kappa$ form, which determines the final GTS model. This significantly reduces training resources when tuning $\kappa$. A more general methodology is left for future work (L341-342).
>
> # Ref
> [1] Deng et al. Calibrating user response predictions in online advertising. 2020
>
> [2] Hothorn et al. Conditional transformation model. 2014
>
> [3] Möst et al. Predicting birth weight with conditionally linear transformation models. 2016
>
> [4] Weng et al. Optdist: Learning optimal distribution for customer lifetime value prediction. 2024

---

> > ### Comment · Reviewer_1oaS · 2025-08-09
> >
> > Thank you to the authors for the response, which has clarified most of my previous concerns.
> >
> > However, my reservations about this paper remain. While I appreciate the authors sharing the industry background of the 'predict-then-rerank' paradigm, this does not fully resolve my concerns. In the rebuttal, the authors claim that end-to-end LTR (Learning-to-Rank) frameworks are 'impractical' in industry, yet the authors have not provided strong evidence from literature or data to support this strong assertion.Furthermore, the authors failed to provide any serious comparison with this cutting-edge paradigm, either through theoretical proofs or experimental results demonstrating the shortcomings of the end-to-end approach relative to their method.

---

> ### Author Response · Authors · 2025-08-04
> **Follow-up Regarding Rebuttal**
>
> Dear Reviewer,
>
> We hope this message finds you well.
>
> Thank you very much for your thoughtful and thorough review of our submission. We truly appreciate the time and effort you have dedicated to providing such constructive feedback.
>
> During the rebuttal period, **we've made every effort to address all of your concerns and questions in detail**. We also incorporated your suggestions, which we believe have helped to further improve the quality and clarity of our paper. If there is **anything else that requires clarification**, we would be more than happy to provide further details.
>
> We completely understand that your schedule is very busy, and we are grateful for the important service you provide to the community. If you have an opportunity to take another look at our responses, we would greatly appreciate any additional feedback or thoughts you might have.
>
> Thank you again for your time and consideration. We look forward to any further comments you may have.
>
> Best regards,
>
> All authors of paper 5113

---

> ### Author Response · Authors · 2025-08-07
> **Follow-up Regarding Rebuttal [Only 53 hours left]**
>
> Dear Reviewer 1oaS,
>
> We hope this message finds you well.
>
> Thank you very much for your thoughtful and thorough review of our submission. We truly appreciate the time and effort you have dedicated to providing such constructive feedback.
>
> As the discussion phase is approaching its end, we would like to kindly **follow up regarding our responses** to your concerns and questions. In our rebuttal **we've made every effort to address all of your concerns and questions in detail**. We also incorporated your suggestions, which we believe have helped to further improve the quality and clarity of our paper. If there is **anything else that requires clarification**, we would be more than happy to provide further details.
>
> We completely understand that your schedule is very busy, and we are grateful for the important service you provide to the community. If you have an opportunity to take another look at our responses, we would greatly appreciate any additional feedback or thoughts you might have.
>
> Thank you again for your time and consideration. We look forward to any further comments you may have.
>
> Best regards,
>
> All authors of paper 5113

---

> ### Author Response · Authors · 2025-08-09
>
> Thank you very very much for your thoughtful response and for raising these critical points. We sincerely apologize for any confusion and lack of rigorous evidence caused by our statement regarding "the impracticality of end-to-end LTR frameworks in industry".
>
> To clarify, our initial assertion about the current impracticality of e2e LTR in industry was based solely on the industrial experience: As far as we are aware, most large-scale recommender systems—including those at TikTok, Kuaishou, and Alibaba—have not yet successfully switched from a multi-stage paradigm (recall-match-rank-rerank) into a single end-to-end LTR paradigm, primarily due to the significant real-world engineering challenges we mentioned previously. However, since we do not have direct literature or quantitative evidence to definitively support this claim, **we respectfully retract our previous statement regarding the "impracticality of end-to-end LTR in industry" and apologize for the caused confusion**. Acutually, we fully acknowledge and believe that end-to-end LTR is a very promising direction of the future of large-scale industrial recommender systems due to its elegant and superior theoretical properties, and we will also keep an eye on this direction for further research.
>
> At the same time, we would like to **maintain our position on the practical significance and value of the "predict-then-rank" paradigm, the research problem on point accuracy, and the contributions of our proposed approach**,which we believe is **the major claim that addresses your core concerns.**
>
> Regarding your suggestion for a rigorous comparison with e2e LTR methods, we feel that such an extensive comparison is **beyond the scope of this paper**. Notably, many impactful regression-based studies—such as CREAD[1], TPM[2], D2Q[3], and the recent GR[4]—have not provided such direct comparisons either. And since we have retracted our claims regarding LTR, it also seems to become unnecessary to compare with it. **We look forward to the discussion ultimately coming to the initial question**: Whether the "predict-then-rank" paradigm and the research problem on point accuracy is significant and valuable—which we believe has been addressed substantively in our rebuttal and in this reply. Inspired by your thoughtful comments, we will consider adding rigorous comparisons with e2e LTR methods in our subsequent exploration of GTS.
>
> Thank you again for your invaluable feedback and for engaging in this constructive discussion. *We hope that the resolved concerns and the further clarification will be reflected favorably in the final evaluation.*
>
> Best regards,
>
> All authors of paper 5113
>
> # Ref
>
> [1] Sun J, et al. Cread: A classification-restoration framework with error adaptive discretization for watch time prediction in video recommender systems. 2024
>
> [2] Lin X, et al. Tree based progressive regression model for watch-time prediction in short-video recommendation. 2023
>
> [3] Zhan R, et al. Deconfounding duration bias in watch-time prediction for video recommendation. 2022
>
> [4] Ma H, et al. Generative Regression Based Watch Time Prediction for Short-Video Recommendation. 2025

---

### Official Review · Reviewer_Fv4e · 2025-07-12

**Clarity:** 2
**Significance:** 2
**Originality:** 3
**Rating:** 4
**Confidence:** 4

**Summary:**

This paper addresses the retransformation bias problem in regression models, which is especially important in recommender systems where data is highly skewed. The authors propose TranSUN, a preemptive method that eradicates this bias intrinsically during training. TranSUN augments standard model with an auxiliary branch that jointly learns a correction factor, ensuring the final prediction is an unbiased estimate of the target's expectation. The paper further generalizes this idea into the Generalized TranSUN (GTS) framework, which reveals that the core mechanism for unbiasedness is "conditional linearity" and provides a flexible approach to debias various regression models. The method's effectiveness is demonstrated on synthetic and real-world datasets, with a successful large-scale deployment in two recommendation scenarios on e-commerce platform.

**Questions:**

See weakness

**Ethical Concerns:**

["NO or VERY MINOR ethics concerns only"]

**Limitations:**

Yes

**Quality:**

3

**Strengths And Weaknesses:**

### Strengths
1. **Practical Problem:** The paper tackles retransformation bias, a theoretically well-known but practically critical issue in industrial recommender systems that directly impacts the accuracy of predicted values like GMV or user engagement.

2. **Theoretically Sound Method:** The proposed TranSUN method is simple yet effective, and its claim of unbiasedness is supported by theoretical derivations. The generalization to the GTS framework provides a deeper understanding of the solution space.
3. **Comprehensive Evaluation:** The experimental validation is quite comprehensive.
    (1) The use of synthetic data with diverse distributions demonstrates the bias in baselines and the unbiasedness of the proposed method.
    (2) The experiments on public and a large-scale proprietary industrial dataset show consistent and significant improvements in bias-related metrics (TRE, MRE) and often in standard regression metrics (NRMSE, NMAE).
    (3) The successful, large-scale online A/B test with significant GMV lift shows promising real-world impact.

### Weaknesses
1. **Lack of Efficiency Analysis:** The paper introduces an auxiliary model branch, which presumably adds computational overhead. While Table 13 explores architectural choices, it focuses on accuracy, not efficiency.

2. **Limited Discussion on Ranking Performance Trade-off:** The results (e.g., Table 7) show that while the method improves value prediction, it can sometimes lead to a slight decrease in ranking performance (XAUC). The authors acknowledge this as a limitation, but a deeper discussion would be beneficial. For example, why does correcting for value bias sometimes harm the relative ordering of items? The fact that XAUC improves for high-value items on the Indus dataset (Table 8) is an interesting counterpoint that is not fully explored. A more thorough analysis of this trade-off would be valuable.

---

> ### Author Rebuttal · Authors · 2025-07-30
>
> We thank all the reviewers for their time and insightful feedback to help improve our work. We appreciate that the reviewers recognized our research problem as practically valuable and our method as simple, effective, theoretically grounded, and "of broad practical potential", supported by exhaustive validation. We have carefully considered all the comments and addressed them in this rebuttal, which will be incorporated into the revised version. If any further clarification is required, please do not hesitate to contact us. Below we address your questions (Q) and weaknesses (W).
> > ### W1: Lack of efficiency analysis in Tab13
>
> In the table below, we supplemented Tab13 with TranSUN's additional memory (Mem) and computational overhead (MFLOPS), and calculated the relative increase (OH) compared with T-MSE. As shown, the only-sharing-embedding architecture scheme is the most cost-effective, maintaining optimal performance while introducing much little overhead.
>
> | Archs on CIKM16 | Mem (MB) | OH (%) | MFLOPS | OH (%) |
> |---|:---:|:---:|:---:|:---:|
> | 0/1 Emb + 0/3 MLP | 610.46 | 100.00 | 350.86 | 100.00 |
> | **1/1 Emb + 0/3 MLP** | **305.29** | **0.02** | **190.52** | **8.60** |
> | 1/1 Emb + 1/3 MLP | 305.27 | 0.01 | 186.12 | 6.09 |
> | 1/1 Emb + 2/3 MLP | 305.24 | 0.00 | 177.62 | 1.25 |
> | 1/1 Emb + 3/3 MLP | 305.23 | 0.00 | 175.47 | 0.02 |
>
> We also provide additional information on the resource overhead associated with the online TranSUN model in the e-commerce platform's recommendation system, which achieves a **high ROI** measure (with Tab6):
> 1. **Training**: Under the same resources (800 workers), the update iteration (global step) number per second of LogMSE and TranSUN  were 89.2 and 86.8, respectively. Therefore,  applying TranSUN only reduces the training speed by 2.7%, with the training time increased by 2.78%.
> 2. **Inference**: The mean and p99 latency of the LogMSE were 27.8ms and 49.8ms, respectively, while the mean and p99 latency of TranSUN were 28.1ms and 51.6ms, respectively.
> 3. **Deployment**: In the online service with 300 million daily active users, deploying TranSUN did not increase CPU resources and only increased GPU resources by 5%.
>
> > ### W2: Limited discussion on the trade-off between point accuracy and XAUC
>
> Here we elaborate on the cause of the trade-off phenomenon between point accuracy and XAUC, and further highlight the limitations of XAUC. This limitation leads to the inconsistency between the offline XAUC metric and online business metrics (see Tab6), making XAUC no longer suitable for offline assessment. To address this, we supplemented the offline evaluation with two more consistent offline metrics, high-value XAUC (L565-568, Tab 8) and a commonly used ranking metric NDCG, on both of which our model demonstrates superior performance. For clarity, let us start with a concrete case:
>  1. **A case of trade-off**: We first sample two inputs {x1, x2} ~ P(X), and then sample 3 groups of target y from P(Y|x1) and P(Y|x2) respectively, and let the samples be {3,3,3} ~ P(Y|x1) and {8,1,1} ~ P(Y|x2), and let T(x)=log(x). Then:
> 	- The LogMSE prediction is $f(x_1)=\mathrm{T}^{-1}(\mathbb{E}[\mathrm{T}(\mathrm{Y})|x_1])=\exp((\log(3)+\log(3)+\log(3))/3)=3$, $f(x_2)=\mathrm{T}^{-1}(\mathbb{E}[\mathrm{T}(\mathrm{Y})|x_2])=\exp((\log(8)+\log(1)+\log(1))/3)=2$, where $f(x_1)>f(x_2)$ and the bias occurs on the sample X=x2
> 	- The TranSUN/GTS prediction is $f(x_1)=\mathbb{E}[\mathrm{Y}|x_1]=(3+3+3)/3=3, f(x_2)=\mathbb{E}[\mathrm{Y}|x_2]=(1+1+8)/3=10/3$, where $f(x_1)<f(x_2)$
> 	- The descending ranking result of LogMSE is {3, 3, 3, 8, 1, 1}, with XAUC = 0.333, NRMSE = 0.324, and TRE = 4/15, while the result of TranSUN/GTS is {8, 1, 1, 3, 3, 3}, with XAUC = 0.233, NRMSE = 0.301, and TRE = 0.
> 	- It is observed that **a trade-off occurs** between point accuracy (NRMSE, TRE) and XAUC, i.e. TranSUN/GTS achieves better point accuracy (NRMSE, TRE) while XAUC worsens.
>  2. **Root cause of the trade-off:**
> 	 - *Reason for the improvement of high-value XAUC & overall point accuracy metric*: TranSUN/GTS compensates underestimation brought by retransformation bias, resulting in better point and rank accuracy of high-value samples  (e.g., for sample $(x_2,8)$, relative error improves from 0.75 to 0.58, and its rank order moves from 4th to 1st, consistent with results on Indus Top(Tab8)), which significantly improves the overall point accuracy (e.g. improvements in NRMSE & TRE in the case) since these metrics are **value-aware** (i.e. metric calculation aware of absolute value of sample label)
> 	 -  *Reason for the drop of overall XAUC*: Correcting retransformation bias also leads to some hard low-value samples being overestimated (e.g., predictions for the two samples $(x_2,1)$ increase from 2 to 10/3, consistent with Fig3c), which dominates the overall XAUC drop since there are far more hard low-value samples than high-value ones in highly right-skewed data and XAUC metric is **value-agnostic** (i.e. metric calculation only aware of the relative rank order of sample label but not aware to its absolute value)
> 3. **Overall XAUC is inconsistent with online metrics, thus we use high-value XAUC and NDCG**:
> 	- The core goal of a recommender system is to improve online business metrics, while XAUC is merely an intermediate metric. For business target modeled by regression (GMV, watch time, etc.), the event of ranking high-valued samples in the correct position is far more important for online metrics than that of low-valued samples. However, these two events are equally weighted (value-agnostic) in the calculation of XAUC, thereby being dominated by abundant low-valued samples when data are highly right-skewed, which probably contribute only a small portion of online metrics (e.g. 70% of the top low-valued samples contribute only 20% of GMV in Indus). Therefore, overall XAUC is no longer consistent with online metrics
> 	- To address this, we supplemented the offline assessment with a more consistent metric, high-value XAUC (L565-568, Tab8), based on which we achieved improved online results (Tab6). Additionally, NDCG also addresses XAUC's limitation, so we also provide their results. In the table below, NDCG is computed within all samples (NDCG@all) and the top 10% high-value ones (NDCG@10%) respectively. Key observations include: 1) TranSUN performs better in NDCG@all than in XAUC since NDCG is more robust to low-value samples. 2) TranSUN shows superior performance in NDCG@10%, consistent with the reasons described in the 2nd point
>
> | Model | NDCG@all (CIKM16) | NDCG@10%(CIMK16) | NDCG@all (DTMart) | NDCG@10% (DTMart) |
> |---|:---:|:---:|:---:|:---:|
> | MSE | 0.9180 | 0.5201 | 0.9770 | 0.9692 |
> | MAE | 0.9497 | 0.5654 | 0.9867 | 0.9665 |
> | LogMSE | 0.9498 | 0.5517 | 0.9960 | 0.9736 |
> | WLR | 0.9475 | 0.5433 | 0.9693 | 0.9609 |
> | ZILN | 0.9485 | 0.5781 | 0.9963 | 0.9756 |
> | MDME | 0.9391 | 0.5671 | 0.9763 | 0.9639 |
> | TPM | 0.9502 | 0.5888 | 0.9960 | 0.9772 |
> | CREAD | **0.9504** | 0.5881 | 0.9962 | 0.9774 |
> | OptDist | 0.9500 | 0.5852 | 0.9965 | 0.9770 |
> | **LogSUN** | 0.9499 | **0.5904** | **0.9967** | **0.9779** |

---

> ### Author Response · Authors · 2025-08-04
> **Follow-up Regarding Rebuttal**
>
> Dear Reviewer,
>
> We hope this message finds you well.
>
> Thank you very much for your thoughtful and thorough review of our submission. We truly appreciate the time and effort you have dedicated to providing such constructive feedback.
>
> During the rebuttal period, **we've made every effort to address all of your concerns and questions in detail**. We also incorporated your suggestions, which we believe have helped to further improve the quality and clarity of our paper. If there is **anything else that requires clarification**, we would be more than happy to provide further details.
>
> We completely understand that your schedule is very busy, and we are grateful for the important service you provide to the community. If you have an opportunity to take another look at our responses, we would greatly appreciate any additional feedback or thoughts you might have.
>
> Thank you again for your time and consideration. We look forward to any further comments you may have.
>
> Best regards,
>
> All authors of paper 5113

---

> ### Author Response · Authors · 2025-08-07
> **Follow-up Regarding Rebuttal [Only 53 hours left]**
>
> Dear Reviewer Fv4e,
>
> We hope this message finds you well.
>
> Thank you very much for your thoughtful and thorough review of our submission. We truly appreciate the time and effort you have dedicated to providing such constructive feedback.
>
> As the discussion phase is approaching its end, we would like to kindly **follow up regarding our responses** to your concerns and questions. In our rebuttal **we've made every effort to address all of your concerns and questions in detail**. We also incorporated your suggestions, which we believe have helped to further improve the quality and clarity of our paper. If there is **anything else that requires clarification**, we would be more than happy to provide further details.
>
> We completely understand that your schedule is very busy, and we are grateful for the important service you provide to the community. If you have an opportunity to take another look at our responses, we would greatly appreciate any additional feedback or thoughts you might have.
>
> Thank you again for your time and consideration. We look forward to any further comments you may have.
>
> Best regards,
>
> All authors of paper 5113

---

### Official Review · Reviewer_mPVQ · 2025-07-16

**Clarity:** 3
**Significance:** 2
**Originality:** 3
**Rating:** 3
**Confidence:** 3

**Summary:**

TranSUN introduces a preemptive retransformation bias correction paradigm by jointly training a primary branch that predicts the transformed target and an auxiliary branch that learns a corrective factor, thereby achieving theoretically guaranteed unbiased regression with strong empirical convergence. The approach is further generalized into the GTS framework, enabling plug-and-play loss functions and correction schemes, and demonstrates substantial improvements across synthetic benchmarks, public datasets, and large-scale industrial A/B tests.

**Questions:**

The authors observe that while TranSUN improves outcome regression accuracy, it may degrade ranking performance, a phenomenon worthy of deeper investigation. In simplified ranking scenarios (e.g., pairwise comparisons), this reduces to a binary decision task, and prior work has shown that more accurate outcome regression does not necessarily lead to better decision quality. Could the authors elaborate on this point?

**Ethical Concerns:**

["NO or VERY MINOR ethics concerns only"]

**Final Justification:**

TranSUN tackles an important topic and shows promising empirical results, but key concerns remain. The claimed convergence is only demonstrated empirically, with no formal theoretical guarantees. The unbiasedness relies on i.i.d. assumptions and accurate correction factor estimation, both of which are not sufficiently addressed and may not hold in practice. These theoretical gaps limit the reliability and generalizability of the method.
Therefore, I do not recommend acceptance.

**Limitations:**

In the conclusion, the authors acknowledge key limitations, including the lack of evaluation on non-right-skewed distributions and the observed performance drop in ranking metrics such as XAUC.

**Quality:**

2

**Strengths And Weaknesses:**

S1: The research topic is valuable, as preemptive debiasing is indeed more practical than post-hoc correction methods that require substantial computational resources in industrial recommender systems.
S2: The motivation behind TranSUN is clearly articulated, and the method is conceptually simple and easy to implement, yet supported by solid theoretical foundations. Moreover, the authors explicitly state the modeling assumptions and provide proofs for unbiasedness.
S3: Extensive experiments are conducted on synthetic data, public datasets, and large-scale industrial A/B tests, offering solid empirical support for the method's effectiveness.

W1: A key claimed advantage of TranSUN is its strong convergence properties. However, this is only demonstrated empirically, and no formal convergence analysis or theoretical guarantees (e.g., convergence rate bounds) are provided.
W2: The theoretical guarantee of unbiasedness implicitly relies on the assumption that training and test data are independently and identically distributed (i.i.d.), which is not explicitly stated. In practice, if distribution shift occurs at deployment, the unbiasedness claim may no longer hold.
W3: Beyond the i.i.d. assumption, achieving unbiasedness requires that the auxiliary model accurately estimates the correction factor (slope model). This implicitly assumes that the learned slope approximates the true ground-truth ratio, which is itself a non-trivial task.

---

> ### Author Rebuttal · Authors · 2025-07-30
>
> We thank all the reviewers for their time and insightful feedback to help improve our work. We appreciate that the reviewers recognized our research problem as practically valuable and our method as simple, effective, theoretically grounded, and "of broad practical potential", supported by exhaustive validation. We have carefully considered all the comments and addressed them in this rebuttal, which will be incorporated into the revised version. If any further clarification is required, please do not hesitate to contact us. Below we address your questions (Q) and weaknesses (W).
> > ### Q1: The reason for the trade-off between point accuracy and XAUC
>
> Here we elaborate on the cause of the trade-off phenomenon between point accuracy and XAUC, and further highlight the limitations of XAUC. This limitation leads to the inconsistency between the offline XAUC metric and online business metrics (see Tab6), making XAUC no longer suitable for offline assessment. To address this, we supplemented offline evaluation with two more consistent offline metrics, high-value XAUC (L565-568, Tab 8) and a commonly used ranking metric NDCG, on both of which our model demonstrates superior performance. For clarity, let us start with a concrete case:
>  1. **A case of trade-off**: We first sample two inputs {x1, x2} ~ P(X), and then sample 3 groups of target y from P(Y|x1) and P(Y|x2) respectively, and let the samples be {3,3,3} ~ P(Y|x1) and {8,1,1} ~ P(Y|x2), and let T(x)=log(x). Then:
> 	- The LogMSE prediction is $f(x_1)=\mathrm{T}^{-1}(\mathbb{E}[\mathrm{T}(\mathrm{Y})|x_1])=\exp((\log(3)+\log(3)+\log(3))/3)=3$, $f(x_2)=\mathrm{T}^{-1}(\mathbb{E}[\mathrm{T}(\mathrm{Y})|x_2])=\exp((\log(8)+\log(1)+\log(1))/3)=2$, where $f(x_1)>f(x_2)$ and the bias occurs on the sample X=x2
> 	- The TranSUN/GTS prediction is $f(x_1)=\mathbb{E}[\mathrm{Y}|x_1]=(3+3+3)/3=3, f(x_2)=\mathbb{E}[\mathrm{Y}|x_2]=(1+1+8)/3=10/3$, where $f(x_1)<f(x_2)$
> 	- The descending ranking result of LogMSE is {3, 3, 3, 8, 1, 1}, with XAUC = 0.333, NRMSE = 0.324, and TRE = 4/15, while the result of TranSUN/GTS is {8, 1, 1, 3, 3, 3}, with XAUC = 0.233, NRMSE = 0.301, and TRE = 0.
> 	- It is observed that **a trade-off occurs** between point accuracy (NRMSE, TRE) and XAUC, i.e. TranSUN/GTS achieves better point accuracy (NRMSE, TRE) while XAUC worsens.
>  2. **Root cause of the trade-off:**
> 	 - TranSUN/GTS compensates underestimation brought by retransformation bias, resulting in better point and rank accuracy of high-value samples  (e.g., for sample $(x_2,8)$, relative error improves from 0.75 to 0.58, and its rank order moves from 4th to 1st, consistent with results on Indus Top(Tab8)), which significantly improves the overall point accuracy (e.g. improvements in NRMSE & TRE in the case) since these metrics are **value-aware** (i.e. metric calculation aware of absolute value of sample label)
> 	 -  Correcting retransformation bias also leads to some hard low-value samples being overestimated (e.g., predictions for the two samples $(x_2,1)$ increase from 2 to 10/3, consistent with Fig3c), which dominates the overall XAUC drop since there are far more hard low-value samples than high-value ones in highly right-skewed data and XAUC metric is **value-agnostic** (i.e. metric calculation only aware of the relative rank order of sample label but not aware to its absolute value)
> 3. **Overall XAUC is inconsistent with online metrics, thus we use high-value XAUC and NDCG**:
> 	- The core goal of a recommender system is to improve online business metrics, while XAUC is merely an intermediate metric. For business target modeled by regression (GMV, watch time, etc.), the event of ranking high-valued samples in the correct position is far more important for online metrics than that of low-valued samples. However, these two events are equally weighted (value-agnostic) in the calculation of XAUC, thereby being dominated by abundant low-valued samples when data are highly right-skewed, which probably contribute only a small portion of online metrics (e.g. 70% of the top low-valued samples contribute only 20% of GMV in Indus). Therefore, overall XAUC is no longer consistent with online metrics
> 	- To address this, we supplemented the offline assessment with a more consistent metric, high-value XAUC (L565-568, Tab8), based on which we achieved improved online results (Tab6). Additionally, NDCG also addresses XAUC's limitation, so we also provide their results. In the table below, NDCG is computed within all samples (NDCG@all) and the top 10% high-value ones (NDCG@10%) respectively. Key observations include: 1) TranSUN performs better in NDCG@all than in XAUC since NDCG is more robust to low-value samples. 2) TranSUN shows superior performance in NDCG@10%, consistent with the reasons described in the 2nd point
>
> | Model | NDCG@all (CIKM16) | NDCG@10%(CIMK16) | NDCG@all (DTMart) | NDCG@10% (DTMart) |
> |---|:---:|:---:|:---:|:---:|
> | MSE | 0.9180 | 0.5201 | 0.9770 | 0.9692 |
> | MAE | 0.9497 | 0.5654 | 0.9867 | 0.9665 |
> | LogMSE | 0.9498 | 0.5517 | 0.9960 | 0.9736 |
> | WLR | 0.9475 | 0.5433 | 0.9693 | 0.9609 |
> | ZILN | 0.9485 | 0.5781 | 0.9963 | 0.9756 |
> | MDME | 0.9391 | 0.5671 | 0.9763 | 0.9639 |
> | TPM | 0.9502 | 0.5888 | 0.9960 | 0.9772 |
> | CREAD | **0.9504** | 0.5881 | 0.9962 | 0.9774 |
> | OptDist | 0.9500 | 0.5852 | 0.9965 | 0.9770 |
> | **LogSUN** | 0.9499 | **0.5904** | **0.9967** | **0.9779** |
>
> > ### W1: Lack of convergence analysis for $L_{sun}^{\mathrm{T}}$
>
> We add the convergence analysis:
> 1. For convergence rate bound, T-MSE (Eq2) = $L_{sun}^{\mathrm{T}}$ (Eq4) = MSE: Under the Lipschitz assumption, since the loss formulations of $L_{sun}^{\mathrm{T}}$, MSE, and T-MSE are all squared errors, the theoretical convergence rate bound using SGD is $O(\frac{1}{\sqrt{K}})$, where $K$ is update iteration number, according to [1].
> 2. **For convergence smoothness, T-MSE > L_{sun} > MSE**: The key difference in the convergence performance of $L_{sun}^{\mathrm{T}}$, MSE, and T-MSE lies in their smoothness. This is because the variance and the outlier proportion of the target are different, resulting in optimization landscapes with different degrees of smoothness. The table below shows the target variance (Var), the variance divided by squared expectation (V/E2), and the outlier proportion (Outlier%) outside of 3 times the standard deviation when T(x)=log(x). As shown, the target of $L_{sun}^{\mathrm{T}}$ has a smaller variance and a better outlier distribution than the vanilla MSE, thus $L_{sun}^{\mathrm{T}}$ converges more smoothly (Fig1). Therefore, TranSUN/GTS essentially decomposes a hard task into two easier and smoother tasks
>
> | Loss (CIKM16) | Var | V/E2 | Outlier% |
> |---|:---:|:---:|:---:|
> | MSE | 16440 | 0.41 | 1.03% |
> | $L_{sun}^{\mathrm{T}}$ | 0.63 | 0.36 | 0.59% |
> | LogMSE | 0.62 | 0.02 | 0.29% |
>
> | Loss (Indus) | Var | V/E2 | Outlier% |
> |---|---|---|---|
> | MSE | 97494.16 | 12.10 | 0.78% |
> | $L_{sun}^{\mathrm{T}}$ | 2.327 | 0.93 | 0.32% |
> | LogMSE | 1.87 | 0.15 | 0.28% |
>
> > ### W2: Unbiasedness relies on I.I.D. assumption
>
> We beg to differ.
> 1. Theoretical unbiasedness in our paper defines that the model prediction f(x0) for any training sample x0 is equal to the conditional expectation E(Y|x0) under the distribution of the training data (Eq13). So our unbiasedness is completely independent to the distribution of the test data (e.g. whether the training-test distribution violates IID).
> 2. IID violation is a problem that neither TranSUN nor any existing post-hoc debiasing and calibration methods can address, and is completely beyond the scope of this paper.
>
> We will clarify this in the revised version.
>
> > ### W3: Unbiasedness requires that the ratio estimation is accurate, which is non-trivial
>
> One point needs clarification: theoretical unbiasedness is an intrinsic theoretical property of TranSUN/GTS and is unaffected by model convergence (L164-165). An example is that the theoretically unbiased MSE fails to converge when modeling high-skewed data, exhibiting quasi-biased performance (Fig1, Tab7). Therefore, I understand your underlying concern is the difficulty of learning the non-trivial ratio estimation task. Regarding this:
>
>  1. **The task is easier** for model than the vanilla MSE, since $L_{sun}^{\mathrm{T}}$ can converge much smoother than the vanilla task (according to the reply to W1), which was also carefully selected from three bias modeling schemes (Tab5, Fig3a&b)
>  2. The comparisons with post-hoc debiasing and calibration methods (Tab4) show that the model is **indeed capable of grasping this task**
>
> We will clarify this in the revised version.
>
> # Ref
>
> [1] Bottou L et al. Optimization methods for large-scale machine learning. 2018

---

> ### Author Response · Authors · 2025-08-05
> **Follow-up Regarding Rebuttal**
>
> Dear Reviewer,
>
> We hope this message finds you well.
>
> Thank you very much for your thoughtful and thorough review of our submission. We truly appreciate the time and effort you have dedicated to providing such constructive feedback.
>
> During the rebuttal period, **we've made every effort to address all of your concerns and questions in detail**. We also incorporated your suggestions, which we believe have helped to further improve the quality and clarity of our paper. If there is **anything else that requires clarification**, we would be more than happy to provide further details.
>
> We completely understand that your schedule is very busy, and we are grateful for the important service you provide to the community. If you have an opportunity to take another look at our responses, we would greatly appreciate any additional feedback or thoughts you might have.
>
> Thank you again for your time and consideration. We look forward to any further comments you may have.
>
> Best regards,
>
> All authors of paper 5113

---

> ### Author Response · Authors · 2025-08-07
> **Follow-up Regarding Rebuttal [Only 53 hours left]**
>
> Dear Reviewer mPVQ,
>
> We hope this message finds you well.
>
> Thank you very much for your thoughtful and thorough review of our submission. We truly appreciate the time and effort you have dedicated to providing such constructive feedback.
>
> As the discussion phase is approaching its end, we would like to kindly **follow up regarding our responses** to your concerns and questions. In our rebuttal **we've made every effort to address all of your concerns and questions in detail**. We also incorporated your suggestions, which we believe have helped to further improve the quality and clarity of our paper. If there is **anything else that requires clarification**, we would be more than happy to provide further details.
>
> We completely understand that your schedule is very busy, and we are grateful for the important service you provide to the community. If you have an opportunity to take another look at our responses, we would greatly appreciate any additional feedback or thoughts you might have.
>
> Thank you again for your time and consideration. We look forward to any further comments you may have.
>
> Best regards,
>
> All authors of paper 5113

---

### Note · Authors · 2025-08-15

We sincerely appreciate the AC and reviewers for your thoughtful reviews and constructive engagement! Please see our final remarks below.

### **Strength Summary**

We appreciate the reviewers’ recognition:

1. Addressing an "interesting"$^{[4]}$, "valuable"$^{[1]}$, and "practically critical"$^{[2,3]}$ problem
2. Proposing a "simple yet effective"$^{[1,2]}$ method (TranSUN) with "solid theoretical foundations"$^{[1,2]}$, with its generalization (GTS) offering "a deeper understanding"$^{[2]}$, which is recognized of "broad practical potential"$^{[3]}$
3. Providing "quite comprehensive"$^{[2]}$ and "very solid"$^{[1,3]}$ empirical validation—spanning synthetic, public, and industrial data, and large-scale online A/B tests—showing "promising results"$^{[4]}$
4. The paper being "clearly written"$^{[4]}$, and motivation"clearly articulated"$^{[1]}$

[1], [2], [3], [4]: Reviewer mPVQ, Fv4e, 1oaS, and xLUq, respectively

### **Rebuttal and Discussion Overview**

In the rebuttal, we addressed all reviewer concerns and questions with added experiments, theoretical evidence, practical details, and illustrative cases. We received two *positive* feedbacks:

- Reviewer xLUq: All concerns well resolved; intended to improve rating
- Reviewer 1oaS: Most concerns clarified and valued our industry perspective; only the ranking paradigm concern ("Predict-Then-Rank" vs. "Learn-To-Rank") remained (which we further addressed; no reply)

Key responses to major concerns:

1. **XAUC vs. point accuracy trade-off**$^{[1,2,3,4]}$: We use a case to illustrate the underlying reason and highlight XAUC's limitation (i.e. inconsistency with online business metric). To address it, high-value XAUC & NDCG are adopted, where our method excels

2. **Convergence of $L_{sun}^{T}$**$^{[1,4]}$: Added experiments show $L_{sun}^{T}$ reduces variance & outliers, smoothing optimization landscape to improve convergence

3. **Efficiency & overhead**$^{[2,3]}$: Added analysis shows our method is efficient both offline & online, delivering high ROI

4. **Under limited PTR paradigm**$^{[3]}$: We support PTR's high practical value with recent impactful related research and the fact that most large-scale recommender systems still use PTR

### **Revision Commitment**

We will carefully incorporate all additional evidence, clarifications, and insights from the rebuttal phase into our revised manuscript to maximize rigor and clarity.

Thank you again for your time, invaluable feedback, and support!

---

### Decision · Program_Chairs · 2025-09-17

**Decision:**

Accept (poster)

**Comment:**

This paper proposes TranSUN, a preemptive paradigm to intrinsically eradicate retransformation bias in regression models for recommender systems, along with its generalization (GTS), backed by solid theoretical guarantees, comprehensive experiments across synthetic, public, and industrial data, and successful deployment in real-world e-commerce scenarios.

The reviews highlight the work’s strengths: addressing a valuable and practically critical problem, with a simple yet effective method supported by strong theoretical foundations and extensive empirical validation. Key concerns raised by reviewers included convergence analysis of $L_{sun}^{T}$, the trade-off between point accuracy and ranking metrics (e.g., XAUC), efficiency overhead, reliance on the "predict-then-rank" paradigm, and comparisons with probabilistic modeling methods. After reading the rebuttal and discussions, I think the authors adequately addressed these concerns by supplementing theoretical analysis and experimental evidence. Although some extra concerns are raised during the discussion period, they are not highly related to the core novelty and scientific soundness of the paper. Overall, I recommend that this paper be accepted.